



# Toward a marginal Arctic sea ice cover: changes to freezing, melting and dynamics

Rebecca Frew[1], Danny Feltham[1], David Schröder[1], and Adam Bateson[1]

[1]Department of Meteorology, University of Reading

**Correspondence:** Rebecca Frew (rebecca.frew@reading.ac.uk)

**Abstract.**

As summer Arctic sea ice extent has retreated, the marginal ice zone (MIZ) has been widening and making up an increasing percentage of the summer sea ice. The MIZ is defined as the region of the ice cover that is influenced by waves, and for convenience here is defined as the region of the ice cover between ice concentrations (area fractions) A of 15 to 80%. The MIZ is projected to become a larger percentage of the summer ice cover, as the Arctic transitions to ice free summers. We compare individual processes of ice volume gain and loss in the ice pack (A>80%) to those in the MIZ to establish and contrast their relative importance and examine how these processes change as the summer MIZ fraction and amplitude of the seasonal sea ice growth/melt cycle increases over decadal timescales. We use an atmosphere-forced, physics-rich sea ice-mixed layer model that includes a prognostic floe size distribution (FSD) model including brittle fracture and form drag. The model has been compared to FSD observations, satellite observation of sea ice extent and PIOMAS.

The MIZ fraction of the July sea ice cover, when the MIZ is at its maximum extent, increases by a factor of 2 to 3, from 14% (20%) in the 1980s to 46% (50%) in the 2010s in NCEP (HadGEM2-ES) atmosphere-forced simulations. In a HadGEM2-ES forced projection the July sea ice cover is almost entirely MIZ (93%) in the 2040s. Basal melting accounted for the largest proportion of melt in regions of pack ice and MIZ for all time periods. During the historical period, top melt was the next largest melt term in pack ice, but in the MIZ top melt and lateral melt were comparable. This is due to a relative increase of lateral melting and a relative reduction of top melting by a factor of 2 in the MIZ compared to the pack ice. The volume fluxes due to dynamic processes decreases due to the reduction in ice volume in both the MIZ and pack ice. As the ice cover becomes marginal (MIZ), it melts earlier: in the region that was pack ice in the 1980s and became marginal in the 2010s, peak melting starts 20/12 days earlier (NCEP/HadGEM2-ES). This continues in the projection where melting in the region that becomes MIZ in the 2040s shifts 14 days earlier.

## 1 Introduction

The marginal ice zone (MIZ) is traditionally defined as the region of the sea ice cover influenced by ocean waves (Dumont et al., 2011; Horvat et al., 2020). Here however, we have applied the often used and more easily applied definition of the MIZ as the region covered by 15-80% sea ice concentration (Rolph et al., 2020; Aksenov et al., 2017; Strong and Rigor, 2013). The pack ice if defined as the region where the ice concentration exceeds 80%. The MIZ grows in early summer as the sea ice cover



starts to melt and become more fragile. This leads to an increase in fragmentation creating a higher concentration of smaller floes. As the sea ice cover shrinks to its minimum extent in the MIZ contracts too. The MIZ forms a much smaller fraction of the sea ice cover throughout the winter months. As summer Arctic sea ice has retreated over the past 40 years the MIZ fraction of the summer Arctic sea ice cover has been increasing (Rolph et al., 2020). This trend is projected to continue (Strong and

Rigor, 2013; Aksenov et al., 2017). In this work we consider how the processes of ice gain and loss in the MIZ and the pack ice differ, and what this may mean for the future Arctic sea ice cover.

The sea ice concentration budget from observations has been constructed for the Arctic by Holland and Kimura (2016) using AMRS-E satellite observations spanning 2003-2010, comparing the relative roles of thermodynamic and dynamic processes through the seasonal cycle. There is no equivalent for ice thickness and volume using observations yet, but a number of studies

have assessed the Arctic sea ice mass budget in climate models (Holland et al., 2010; Keen and Blockley, 2018; Keen et al., 2021). Holland et al. (2010) assessed CMIP3 models and found a large amount of variation between the relative importance of processes as Arctic sea ice declined. They also found that the initial sea ice state was important in determining projected changes to the sea ice cover, with thicker initial ice resulting in more sea ice volume change.

Following the framework set out by the Sea-Ice Model Intercomparison Project (SIMIP) in Notz et al. (2016) for comparing

the energy, mass and freshwater budgets, Keen et al. (2021) compared the sea ice mass balance in CMIP6 models over the 21st century. Although Keen et al. (2021) also found significant differences in the changes to the mass budget component size and timing between the models, they found that when the sea ice state is taken into account the models behave in a similar fashion to warming, with melting happening earlier in the summer and growth reducing in autumn and increasing in winter over the coming decades.

In this study we present an analysis (the first to our knowledge) of the relative contribution of sea ice processes controlling the mass balance in the pack ice and MIZ. This motivates the use of a sea ice model with a higher physical fidelity than used by climate models and able to capture the distinction of MIZ processes. We use a local (CPOM) version of the CICE sea ice model coupled to a mixed layer model (Petty et al., 2014), which is forced by atmospheric reanalysis and an ocean climatology. This model includes various refinements to the physics (Schröder et al., 2019) and in particular includes a prognostic floe distribution

(FSD) model, which is important in realistically representing processes in the MIZ where there is a higher concentration of smaller ice floes. We use a FSD model based on Roach et al. (2018, 2019) that includes brittle fracture (Bateson et al., 2022), found to give realistic simulations of observed FSD for mid-range floe sizes in the Arctic. Using a forced sea ice-mixed layer model does mean that there will be no feedbacks to the atmosphere. There will also be no impact of trends in oceanic properties, such as the "Atlantification" of the Arctic as the subsurface Atlantic Water layer becomes warmer and thicker (Grabon et al.,

2021), which has the potential to cause sea ice loss if the heat reaches the surface (Carmack et al., 2015; Onarheim et al., 2014; Polyakov et al., 2013). However, field observations indicate that the majority of the ocean heat needed to explain basal ice melt rates can be explained from solar radiation (Perovich et al., 2011), something our model will suitably capture. Note that using coupled and climate models introduces its own set of problems, e.g. CMIP6 models fail to simulate a realistic seasonal cycle of sea ice area (Notz and Community, 2020). Using a forced sea ice model allows us to simulate a more realistic sea ice state,

which has been shown to affect the balance of sea ice processes (Holland et al., 2010; Keen et al., 2021).



In this study we carry out comparisons between different time periods. Between the 1980s where there is a relatively small summer MIZ and the 2010s where there is a large summer MIZ. Then between the 2010s and the 2040s, when the summer sea ice cover is almost entirely MIZ. We quantify the ice volume budget in the different periods, in particular studying the region that becomes MIZ in each of the comparisons to investigate the changes in the volume budget as the MIZ expands. The terms of the sea ice volume budget we examine are:

- **Congelation** growth - basal thickening of the sea ice which occurs from autumn until spring

- **Frazil** ice formation - supercooled seawater freezes to form frazil crystals which clump together to create sea ice

- **Snowice** - snow ice formed when the snow layer on top of the sea ice is pushed below water, is flooded and freezes

- **Basal melting** - melting at the base of the sea ice

- **Top melting** - melting on the surface of the sea ice, which may form melt ponds

- **Lateral melting** - melting at the edge of the sea ice floes

- **Sublimation** - sublimation from the surface of the sea ice. In this study this term includes snow sublimation

- **Dynamics** - the sum of advection, convergence and ridging. This redistributes the ice and can cause either loss or gain of sea ice in a given region. This occurs all year round

The structure of this paper is as follows. The sea ice-mixed layer model is described in Section 2.1. The NCEP and HadGEM2-ES forced simulations are compared to satellite observations and PIOMAS ice volume estimates in Section 3. The ice volume fluxes in the pack ice and MIZ in low MIZ (1980s), high MIZ (2010s) and all MIZ (2040s) scenario are compared in Section 4. The total annual fluxes are shown in Section 4.1 followed by the annual cycle in the main melt and growth terms in Section 4.2. Finally the main findings are summarised in the Concluding remarks in Section 5.

## 2 Methodology

### 2.1 Model set up

We use a dynamic-thermodynamic sea ice model, CICE coupled to a mixed layer model of Petty et al. (2014). We use the CPOM version of CICE which is based on version 5.1.2 (Hunke et al., 2015). This version has been calibrated to Cryosat-2 data (Schröder et al., 2019). This version of CICE uses the form drag scheme of Tsamados et al. (2014) and a prognostic floe size distribution (FSD) model (Roach et al., 2018, 2019) that includes a parameterisation of brittle fracture (Bateson et al., 2022). The addition of this brittle fracture parameterisation improves model performance in simulating the shape of the FSD for mid-sized floes. Examples of previous use of the CPOM version of the CICE model are demonstrated in Rolph et al. (2020) without the addition of brittle fracture in the prognostic FSD model, and in Schröder et al. (2019) without a prognostic FSD model or prognostic mixed layer model.



We run the model in standalone mode for the pan-Arctic, with a grid resolution of ∼40 km. The ocean temperature and salinity below the mixed layer are restored to climatological means from MYO-WP4-PUM-GLOBALREANALYSIS-PHYS-001-004 (Ferry et al., 2011) over a time scale of 20 days. The ocean currents are restored to values from the same dataset also over 20 days. The mixed layer temperature, salinity and depth are calculated based on heat and salt fluxes from the deeper ocean and the atmosphere/ice at the surface. We use a number of the default CICE settings including seven vertical ice layers,

one snow layer, thermodynamics of Bitz and Lipscomb (1999), Maykut and Untersteiner (1971) conductivity, the Rothrock (1975) ridging scheme with a Cf value of 12 (an empirical parameter that accounts for dissipation of frictional energy), the delta-Eddington radiation scheme (Briegleb and Light, 2007), and the linear remapping ice thickness distribution (ITD) approximation (Lipscomb and Hunke, 2004). Additionally we use a prognostic melt pond model (Flocco et al., 2010, 2012), an anisotropic plastic rheology (Heorton et al., 2018; Tsamados et al., 2014; Wilchinsky and Feltham, 2006) and a prognostic floe

size distribution model (Roach et al., 2018) that includes brittle fracture (Bateson et al., 2022).

We use two atmospheric forcing sets, NCEP Reanalysis-2 (NCEP2) (Kanamitsu et al., 2002 (updated 2017) atmospheric forcing from 1979 to 2020 and HadGEM2-ES (RCP8.5) (Jones et al., 2011) forcing from 1980 to 2050. The HadGEM2-ES product is purely model based (no data assimilation) and is included to allow us to consider a projection into the mid twenty first century, which enables us to study changes as the summer sea ice cover becomes entirely MIZ. HadGEM2-ES has been

shown to simulate a realistic Arctic sea ice cover (Wang and Overland, 2012), making it a suitable choice.

## 3   Simulated and observed sea ice extent and volume

Here the sea ice extent and volume from the NCEP and HadGEM2-ES forced simulations are compared alongside extent from satellite observations, in Section 3.1, and ice volume from Pan-Arctic Ice Ocean Modeling and Assimilation System (PIOMAS) (Zhang and Rothrock, 2003), in Section 3.2.

### 3.1   Sea ice and MIZ extent

The NCEP forced simulation spans from 1979 to 2020, whilst the HadGEM2-ES simulation spans from 1980 to 2050. Both were initialised with a 6 year spin up period. The minimum and maximum sea ice extent and MIZ extent for each simulation and period studied is given in Table 1. The values are calculated from the average annual cycle of sea ice/MIZ extent of the last 5 years of each decade in the simulations. The period of 5 years was chosen in order to try to capture particular MIZ states

in the simulations without averaging over larger windows. We chose the last 5 years of the 1980s, 2010s and 2040s to reflect a *low MIZ state*, *high MIZ state* and *all MIZ state* in the simulations. The maximum sea ice extent is the same in both simulations in the 1980s, and similar in the 2010s, showing a relatively modest decline from $1.22 \times 10^7$ km$^2$ in the 1980s to $1.15 \times 10^7$ km$^2$ in the 2040s in the HadGEM2-ES forced simulation. The NCEP forced simulation shows a stronger declining summer sea ice extent trend than the HadGEM2-ES forced one, which can be seen in the sea ice extent minima values. In the 2040s the

minimum sea ice extent goes below the value commonly used to define the Arctic as seasonally ice free ($1 \times 10^6$ km$^2$).



|  | 1980s NCEP | 1980s HadGEM | 2010s NCEP | 2010s HadGEM | 2040s HadGEM |
|---|---|---|---|---|---|
| **Max sie** | $1.22 \times 10^7$ km$^2$ | $1.22 \times 10^7$ km$^2$ | $1.21 \times 10^7$ km$^2$ | $1.19 \times 10^7$km$^2$ | $1.15 \times 10^7$km$^2$ |
| **Min sie** | $6.30 \times 10^6$km$^2$ | $5.94 \times 10^6$km$^2$ | $2.50 \times 10^6$km$^2$ | $2.81 \times 10^6$km$^2$ | $3.15 \times 10^5$km$^2$ |
| **Max MIZ** | $2.03 \times 10^6$km$^2$ | $3.99 \times 10^6$km$^2$ | $4.58 \times 10^6$km$^2$ | $4.75 \times 10^6$km$^2$ | $6.46 \times 10^6$km$^2$ |
| **Min MIZ** | $8.82 \times 10^5$km$^2$ | $7.42 \times 10^5$km$^2$ | $7.96 \times 10^5$km$^2$ | $7.07 \times 10^5$km$^2$ | $1.74 \times 10^5$km$^2$ |

**Table 1.** Table of sea ice extent and MIZ extent minimum and maximum values for each time period. In each case the last 5 years of sea ice extent in each decade has been averaged and the minimum and maximum sea ice extent and MIZ extent is values for sea ice and MIZ extent is calculated from the averaged annual cycle.

In Figure 1 we compare the two simulations to each other and satellite observations from NASA Team and NASA Bootstrap, which we chose to give an upper and lower estimate of MIZ extent values from satellite observations. The Bootstrap and NASA Team sea ice extent estimates are very similar to each other. Generally the sea ice extent is slightly lower in the satellite observations in June and July and slightly higher in August and September. The sea ice extent is generally lower in the NCEP forced simulation than the HadGEM2-ES forced simulation, with lower MIZ extent, especially in August and September. NASA Team gives us an upper estimate of MIZ extent, whilst Bootstrap gives us a lower estimate. The simulations start off generally closer to the lower estimate (Bootstrap) in the summer months with the exception of the HadGEM2-ES forced simulation in August and September. The simulations then tend more towards the upper estimate (NASA Team) in July, August and September months. This is due to the increasing MIZ extent trend present in the simulations which is not in the observations. Despite this discrepancy, the forced simulations fall within the observational estimates which gives us confidence that they are simulating an appropriate sea ice state and are suitable for this study. There is a large range in summer MIZ estimates between different satellite products, as discussed in Rolph et al. (2020). Both simulations are on the lower end of estimates in June, and the upper end in July and August when the MIZ peaks. By the 2010s the MIZ becomes the dominant part of the Arctic sea ice cover in the simulations and observations, and by the end of the 2030s the summer sea ice cover is almost entirely MIZ in the HadGEM2-ES forced simulation.

### 3.2 Sea ice volume

Total sea ice volume over the Arctic ocean (defined here as being ocean North of 66.5°N) is plotted for HadGEM2-ES and NCEP forced simulations alongside monthly values from PIOMAS in Figure 2. The NCEP forced model is relatively similar to the PIOMAS estimate, showing similar variability, but overestimates the seasonal cycle, mostly due to too much sea ice volume in the winter. Meanwhile the HadGEM2-ES forced simulation underestimates the sea ice volume all year round, although it does tend towards to the PIOMAS estimate over time due to showing a smaller sea ice volume decrease over time. Both simulations produce suitably realistic sea ice extent and volume for use in this study.





**Figure 1.** Monthly June, July, August and September sea ice (solid lines) and MIZ extent (dashed lines) from the NCEP (1979-2020) and HadGEM2-ES (1980-2050) forced simulations compared to satellite observations from NASA Team (1979-2020) and NASA Bootstrap (1979-2020). Yellow shaded areas show the three study 5 year periods used in Section 4.





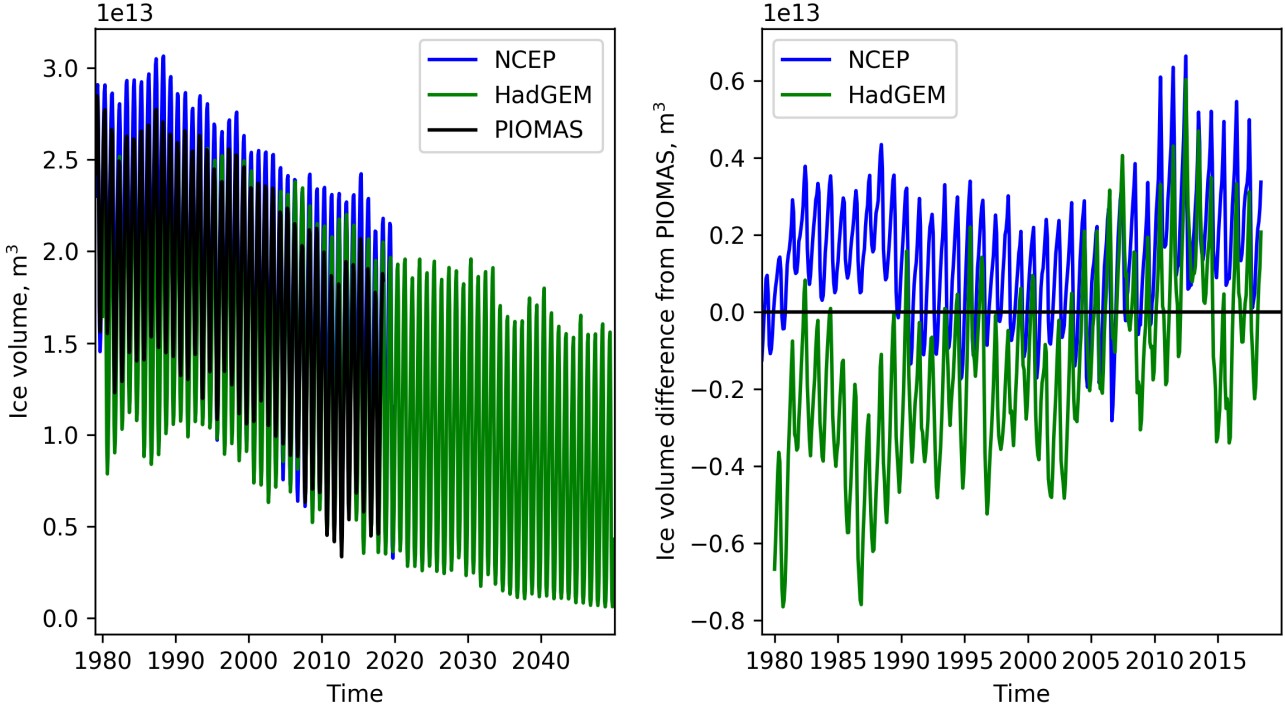

**Figure 2.** Monthly Arctic sea ice volume from NCEP and HadGEM2-ES forced simulations compared to PIOMAS in (a) and the differences from PIOMAS shown in (b).

## 4 Results: Volume fluxes in the pack ice and MIZ

Here we compare the sea ice volume fluxes in the pack ice and MIZ. The MIZ is taken to be between 15-80% sea ice concen-
tration whilst the pack ice is taken to be regions of sea ice concentration above 80%. We consider three different ice cover states
within the simulations: a *low MIZ state* in the 1980s; a *high MIZ state* in the 2010s; and an *all MIZ state* in the 2040s. Figure 3
shows the change in MIZ coverage in the summer based on daily July sea ice concentration fields from the two simulations plus
NASA Team and NASA Bootstrap from the 1980s to the 2010s, and then from the 2010s to the 2040s from the HadGEM2-ES
forced simulation. In each case we use the last 5 years of daily July sea ice concentration and assign each grid cell as pack
ice (ice concentration≥80%), MIZ (15% ≤ice concentration<80%) or open water (ice concentration<15%). We then compute
where a grid cell spends most of its time in each time period to define each region as pack ice,MIZ or open water. This gives a
more accurate representation of where the MIZ is observed and simulated than computing the MIZ from time averaged sea ice
concentration fields. The use of fixed regions for our analysis means that they do not reflect what is MIZ and pack ice on each
day of the year, however it does enable to analyse volume fluxes in the region that is predominantly MIZ in July.
Region 1, *always pack ice* (blue in Figure 3), is the area that was pack ice in both the 1980s and 2010s (2010s and 2040s);
region 2, *becomes MIZ* (green), is the area that was pack ice in the 1980s (2010s) and became MIZ in the 2010s (2040s);



and region 3, *always MIZ* (orange), is the region that was MIZ in both the 1980s and 2010s (2010s and 2040s). There is a large range in the MIZ coverage estimated by satellite products Rolph et al. (2020), we chose Bootstrap and NASA Team for this comparison to give an indication of lower and upper observational estimates of MIZ extent. NASA Team has a much larger region that *becomes MIZ* and is *always MIZ* than Bootstrap. The MIZ coverage in the NCEP and HadGEM2-ES forced simulations is closer to Bootstrap in the 1980s and closer to NASA Team in the 2010s as given in Table 2. The simulations show a different spatial coverage to the MIZ and changes compared to the satellite observations. The simulations show a larger increase in the MIZ around the Fram Strait region, particularly in the HadGEM2-ES forced simulation. There is a similar change in percentage of MIZ coverage in the two simulations, with the HadGEM2-ES forced simulation showing slightly more MIZ in both periods. By the 2010s the MIZ is making up 46%/50% of the sea ice cover in the NCEP/HadGEM2-ES forced simulations, and 93% by the 2040s in the HadGEM2-ES forced projection. We analyse the simulated annual volume fluxes in Section 4.1 and annual cycles for the melt terms and congelation growth in Section 4.2 using the regions defined in Figure 3.

## 4.1 Annual total volume fluxes

Here we discuss the annual volume fluxes for sea ice processes as shown in Figure 4 and the relative importance of top, basal and lateral melting in the 1980s as shown in Figure 5. Congelation growth dominates sea ice growth making up between 91-95% in both pack ice and MIZ regions with no clear change over time or between the pack ice and MIZ. Frazil is the next biggest ice growth term making up between 5-9%. In all cases snow ice formation is a negligible contribution to sea ice growth making up less than 1% in all periods and regions.

The partitioning of the melt between top, basal and lateral melting does differ significantly between the pack ice and MIZ. In all regions and simulations basal growth makes up the largest proportion of melting ranging between 46-60% in regions of pack ice and 52-69% in regions of MIZ across the simulations and time periods. In the 1980s and 2010s simulations top melting is significantly more important in the pack ice than in the MIZ, this is predominantly compensated by an increase in lateral melting in the MIZ alongside an increase in basal melt in the MIZ relative to the pack. Top melt makes up 39-50% in the *always pack* region and 15-21% in the *always MIZ* region in the 1980s and 2010s. Whilst the *becomes MIZ* values are intermediate to the other two regions with top melt making up 30-36%. Lateral melting makes up 4-5% of melting in the *always pack* region, 6-9% in the *becomes MIZ* region and 10-15% in the *always MIZ* region.

It is surprising that there is not a more significant change over time in the *becomes MIZ* region which is pack ice in the 1980s(2010s) and MIZ in the 2010s(2040s) in terms of proportion of melt that is top, basal or lateral. The values are approximately midway between those of the *always pack* and *always MIZ* regions. This likely reflects that the contrasts between the pack ice and the MIZ regions are sensitive to the sea ice concentration gradient and associated properties, as opposed to the binary classification of pack ice or MIZ. This means that it is likely that MIZ closer to the pack ice has a different balance of processes to the outer MIZ that has lower ice concentrations.

The top, basal and lateral melt fractions in the projections do not match the earlier period in the HadGEM2-ES forced simulation. There is much more top melting in both the *always MIZ*, and to a lesser degree the *becomes MIZ* region. This result is consistent with the increase in top melting seen in the near future in CMIP6 model projections (Keen et al., 2021).





The changing location of the sea ice might be a partial explanation of the change, as the sea ice moves to higher latitudes this may have an impact on the balance of processes in addition to whether the sea ice is located in the pack ice or MIZ. A possible explanation for the increase in top melting in the future MIZ simulated could also be the use of constant ocean forcing in the projection, which may be lacking ocean warming and therefore not represent realistic ocean conditions for the projection. This means that the results here could be seen as a lower estimate on the ice loss and changes that we will expect to see in the 2040s depending on the speed and magnitude of warming of Atlantic Waters entering the Arctic Ocean. Feedbacks with the atmosphere may also act to speed up and intensify the changes in ice and temperature that we might expect to see in the 2040s.

Dynamics (advection, convection and ridging) results in a net negative flux in all regions, though the magnitude is significantly larger in the pack ice than in the MIZ as might be expected due to a larger volume of sea ice that can be advected. These results reflect that generally sea ice is exported outwards from the central Arctic and melts at lower latitudes. In all of the regions the magnitude of the negative dynamics sea ice flux decreases over time, reflecting the reduction in sea ice volume. In the NCEP simulation the decrease in negative volume flux in the 2010s relative to the 1980s was relative similar in all of the regions, ranging 31-34%. In the HadGEM2-ES forced simulation more of a contrast across the regions, with a larger decrease in the MIZ likely reflecting larger ice loss. With a decrease of 31% in the *always pack* region, 43% decrease in the *becomes MIZ* region and 51% in the always MIZ region. In the the projection the change from the 2010s to the 2040s we see a continuation of the prior changes, with a 48% decrease in the *becomes MIZ* region and a 41% decrease in the *always MIZ* region.

## 4.2   Annual cycle of the main melt and growth terms

To better understand the net changes in annual sea ice volume fluxes and investigate changes to the onset and length of melting we looked at the average annual cycle of congelation growth, basal, top and lateral melt fluxes, shown in Figure 6. The same regions and time periods defined in Figure 3 are used.

Melting occurs first in the outer regions (*always MIZ* regions) and progresses inwards across the sea ice cover to the *always pack* region. This is more pronounced in the NCEP simulation. Top melting occurs first, followed by basal melting in the *always pack* and *becomes MIZ* regions. Lateral melting has a less pronounced summer peak that appears later in the melt season (early august) than the other melt terms, reflecting the increase in fragmentation of the sea ice cover as the summer progresses. Lateral melting is a larger melt term in the MIZ regions, as documented in the previous section. Melting rates and growth rates (per unit area) are larger in the MIZ than the pack ice in the 1980s in both simulations, this difference decreases in the 2010s as melting and growth fluxes increase in the *always pack* region by more than in the *becomes MIZ* and *always MIZ* regions, reflecting the increase in seasonality in the pack ice.

In the *always pack* region peak melt increases and the melt season gets longer in the 2010s relative to the 1980s by 13 days in the NCEP and 6 days in the HadGEM2-ES forced simulations. This is primarily due to earlier melting onset by 9 days in the NCEP and 8 days in the HadGEM2-ES forced simulation. Peak melting rates increase, particularly in the NCEP simulation where they increase by 49%, compared to a 17% increase in the peak melting rate in the HadGEM2-ES simulation. This is partially compensated by an increase in basal growth rates in the Autumn. The increase particularly large in the NCEP simulation where congelation growth increases by 74% on average over the October, November and December, partially





| Time Period | NCEP | HadGEM | Bootstrap | NASA Team |
|---|---|---|---|---|
| **1980s** | 14% | 20% | 19% | 42% |
| **2010s** | 46% | 50% | 30% | 59% |
| **2040s** | - | 93% | - | - |

**Table 2.** Table of July MIZ fractions for each time period, where MIZ fraction is the fraction of the sea ice extent which is defined as MIZ.

compensating the large increase in melting. The average increase in basal growth rates in the HadGEM2-ES forced simulation is a more modest 17%. Note here we use $2.5 \times 10^{-3} \mathrm{m}^3\mathrm{m}^{-2}\mathrm{day}^{-1}$ as the threshold for melting starting and ending to capture the main phases of melt and growth.

In the *becomes MIZ* region the increase in peak rate of melting is again larger in the NCEP simulation, but not as dramatic as in the *always pack* region. Peak total melting rates increase by 29% in the NCEP and 13% in the HadGEM2-ES forced

simulation and there is a shift in the melting season in the 2010s relative to the 1980s. The start of melting shifts earlier by 9 days in the NCEP and 5 days in the HadGEM2-ES forced simulation, whilst the end of summer melting ends later by 16 days later in the NCEP and 14 days in the HadGEM2-ES forced simulation, lengthening the melt season. Peak melting occurs much earlier by 20 days in the NCEP and 12 days in the HadGEM2-ES forced simulation. The change in growth over October, November and December where the growth rates are typically increasing contrast between the two simulations. In the NCEP

simulation there is a 17% increase, whilst in the HadGEM2-ES simulation there is a 6% decrease, likely due to the larger melt rates in the NCEP simulation.

In the *always MIZ* region the melt peaks stays a similar magnitude and occurs at a similar time, only a few days earlier in both simulations. Melting onset shifts slightly earlier by 6 days in the NCEP and 5 days in the HadGEM2-ES simulation. In the HadGEM2-ES simulation there is a significant shift earlier in the end of melting by 19 days (just 4 days earlier in

the NCEP simulation), reflecting a significant reduction in summer ice in that region in the simulation. Growth rates in both simulations decrease over the October-December period, with the NCEP average decreasing by 16% and the HadGEM2-ES average decreasing by 20%.

In the projection, moving from the 2010s to the 2040s we see the same trends in the *becomes MIZ* and *always MIZ* regions as seen in the 1980s to 2010s comparison. The start of melting shifts earlier by 7 days in both the *becomes MIZ* and *always*

*MIZ region*. The melt season shrinks, mostly due to the large shift on the end of the melt season, by 20 days in the *becomes MIZ* region and 21 days in the *always MIZ* region. This reflects all of the ice in those regions having melted. This is combined with a later start to congelation growth in the Autumn by 9 days in the *becomes MIZ* region and 21 days in the *always MIZ* region, followed by slower growth rates in both regions, 17% and 37% slower in the *becomes MIZ* and *always MIZ* regions over October, November and December.

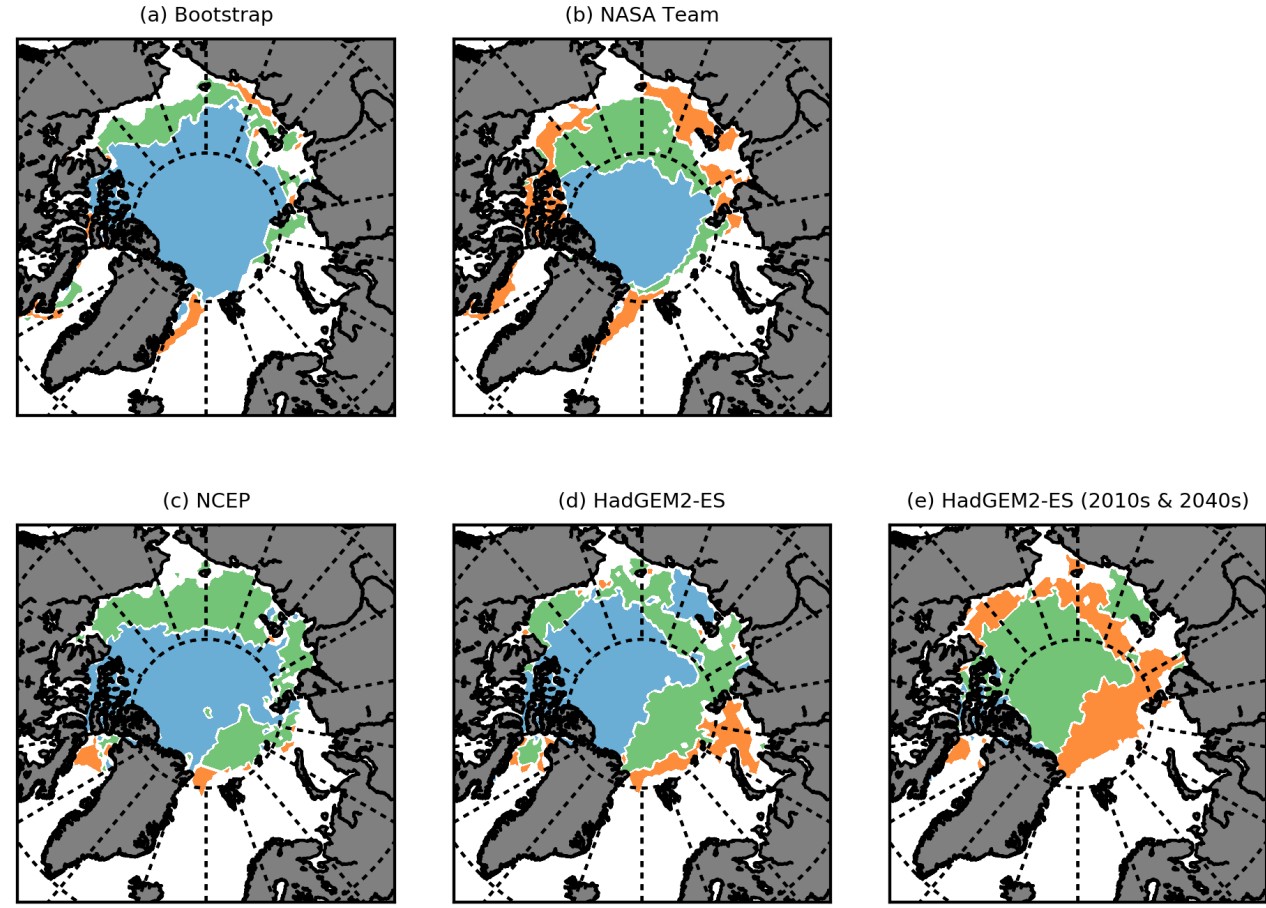

**Figure 3.** Regions of the Arctic sea ice cover defined from daily July ice concentration fields from each time period from the NCEP and HadGEM2-ES forced simulations. Region 1 (blue) indicates the area that is pack ice in both the 1980s(2010s) and 2010s(2040s) in subplots a-d(e). Region 2 (green) indicates the area that is pack ice in the 1980s(2010s) and becomes MIZ in the 2010s(2040s) in subplots a-d(e). Region 3 (orange) indicates the area that is MIZ in both the 1980s(2010s) and the 2010(2040s) in subplots a-d(e).



**Figure 4.** The total annual volume fluxes of sea ice in each of the regions shown in figure 3 for congelation ice growth, frazil ice formation, top melt, basal melt, lateral melt, sublimation, dynamics (transport, convergence and ridging), snow ice formation and the sum of all the terms. The summed annual volume fluxes are calculated from the average annual cycle in each time period. A is the ice area, which is given for each time period and each region and is the annual average value.

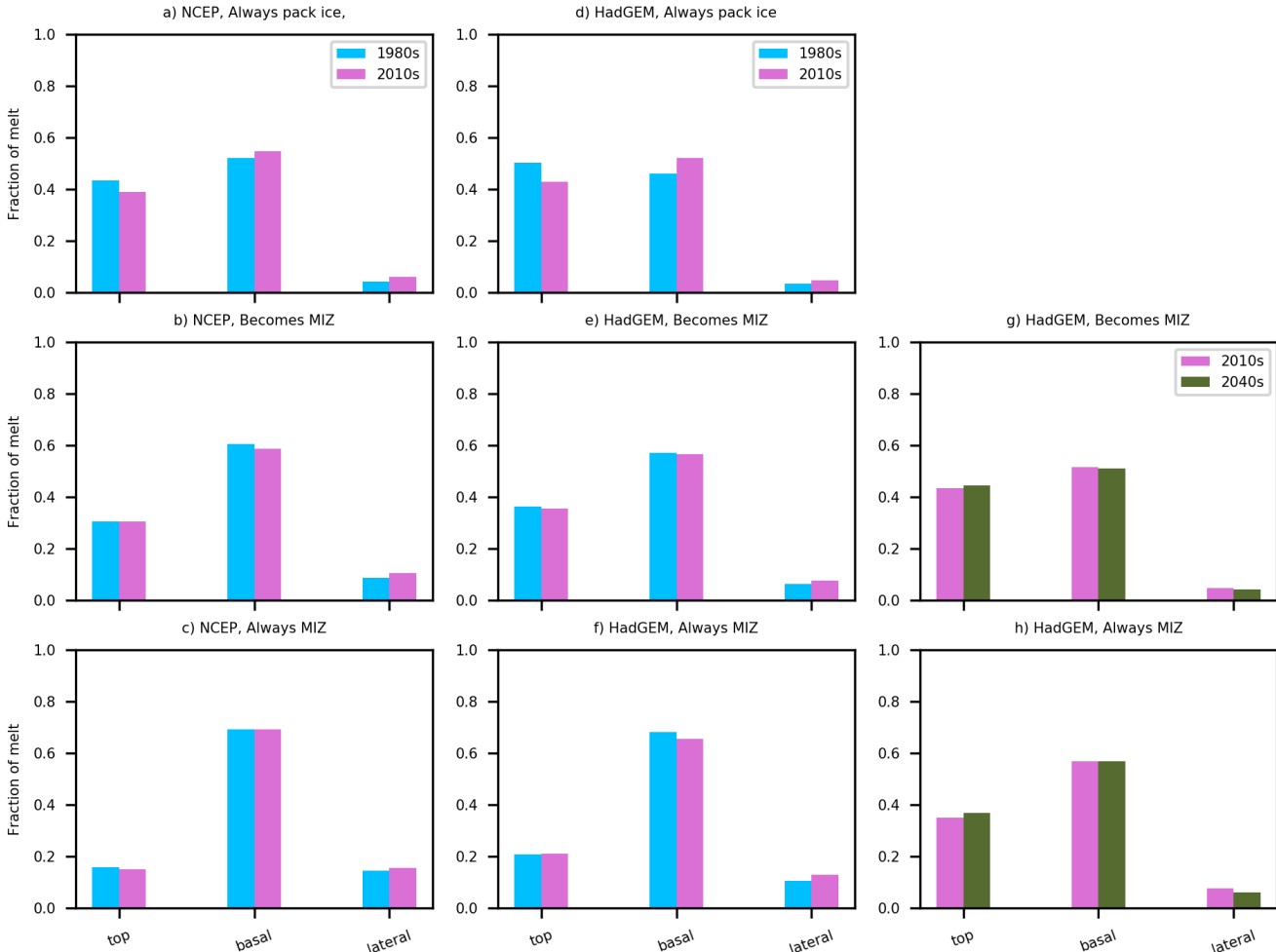

**Figure 5.** Fraction of melt that is top, basal and lateral in each region and time period.

## 5 Concluding remarks

In this study we used a sea ice model coupled to a mixed layer model to compare the ice volume budget in the pack ice and the marginal ice zone (MIZ). The MIZ is defined as having an ice concentration (A) between 15% and 80% and pack ice is defined as A>80%. We ran two simulations, where the model is forced with NCEP reanalysis (1980-2020) and HadGEM2-ES (1980-2050) atmospheric fields. We simulate a MIZ extent within the bounds of observational estimates from Bootstrap and NASA Team, giving us confidence that the model is simulating a realistic sea ice and MIZ state. The NCEP and HadGEM2-ES forced simulations give realistic (and similar) sea ice states over the historical period.

The 1980s *low MIZ* state, and the 2010s *high MIZ* state were compared in simulations using by NCEP and HadGEM2-ES forcing, and the 2010s *high MIZ* state was compared to the 2040s *all MIZ* state. The percentage of the summer sea ice cover



**Figure 6.** The time averaged annual cycles of congelation ice growth, top melt, basal melt and lateral melt in the regions described in Figure 3. The solid line show the 1980s average in subplots (a-f) and the 2010s in subplots (g-h). The dashed lines show the 2010s average in the subplots (a-f) and the 2040s in subplots (g-h).



that was MIZ increased from 14% in the 1980s to 46% in the 2010s in the NCEP forced simulation, and from 20% in the 1980s
to 50% in the 2010s and 93% in the 2040s in the HadGEM2-ES forced simulation.

Sea ice growth was dominated by congelation growth across the pack ice and MIZ regions in all time periods/MIZ states studied, making up between 91-95%. Frazil made up 5-9% of sea ice growth whilst snow ice growth accounted for less than 1% of sea ice growth. There was no significant difference over time or between the pack ice and MIZ in the processes of sea ice growth. Dynamics acted as a volume sink in all regions, as sea ice is transported from the central Arctic to lower latitudes
where it melts. Due to the decreasing sea ice volume this sea ice sink decreased over time in both simulations.

There was a significant contrast in the relative balance of basal, top and lateral melt in the pack ice and MIZ in the 1980s and 2010s in both simulations. Basal melting accounts for the largest portion of melting making up 46-60% in regions of pack ice and increasing slightly in regions of MIZ to 52-69%. Top melt was twice as important in the region that was always pack ice compared to the region that was MIZ in both the 1980s and 2010s, making up 39-50% of melt in the regions that remained pack
ice and 15-21% in the regions that remained MIZ. The opposite is true for lateral melting which makes up 4-5% of melting in the region that remained pack ice and 10-15% in regions that remained MIZ. However, in the projection into the 2040s top melt formed an increased proportion of melting including in the MIZ, which forms most of the sea ice cover during this time period, with values of around 40%.

The timing of the annual seasonal cycles of growth and melt changed significantly in all regions. In the regions that remains
pack ice in the 1980s and 2010s the total melting and growth rates increase, this is more pronounced in the NCEP forced simulation where we see an increase of 49% in the peak total melting rates which is partially compensated by an 74% increase in the average October-December growth rates. In the regions that became or remained MIZ from the 1980s to the 2010s, and for the 2010s and 2040s, we see the melt season shift earlier. Additionally the end of summer melting shifts earlier in all of the regions, particularly in the region that is remains MIZ. This reflects that as the sea ice volume decreases we get to a point
where all of the ice in these regions is melting over the summer, and a period in late summer starts to open up in these regions where there is no sea ice present.

*Author contributions.* RF carried out the calculations and wrote the manuscript. DF and DS gave supervision. AB gave advice on the CICE set up and provided the local CICE version that was used.

*Competing interests.* DF and DS are editors for The Cryosphere.



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
