# Peer review of "Toward a marginal Arctic sea ice cover: changes to freezing, melting and dynamics"

_The Cryosphere, 2023_

## Author Comment (AC2)

*Review 1*

*General comments –*

*The paper provides an assessment of the mass budgets in the MIZ/non-MIZ regions in Arctic sea ice simulations. It uses a comprehensive sea ice model which has been well utilized. It acknowledges that the lack of active atmosphere and dynamic ocean components affects feedbacks in the system, although some deeper discussion of how this might affect the results could be helpful. Overall, I believe that the experiment design is reasonable and can provide some valuable insights on changing Arctic sea ice mass budgets and their projection into the future. However, as noted below, I believe that there is a need for a somewhat deeper analysis and more interpretation of the results.*

*For example, the study explores differences between simulations with different atmospheric forcing. However, it provides little information on why the **NCEP forced vs HadGEM forced runs** simulate different behavior. Additionally, the comparison of MIZ versus non-MIZ mass budgets is given but again there is limited information on **why the mass budgets differ across these different regions**. I do believe that there can be value in delimiting the analysis into MIZ/non-MIZ/changing to MIZ regions. However, I felt that there was a missed opportunity within the manuscript to better articulate **why this approach was useful and what new insight** it provided relative to previous studies on sea ice mass budgets. Additionally, it would have been useful to discuss the **implications from these new insights on broader questions** such as the future evolution of the sea ice and/or discrepancies across models, for example. Overall, I believe that some deeper analysis (including possible new analysis) and deeper discussion of the study implications are needed. With changes in these areas, I believe that the originality and significance of this work would be clearer and this would result in a more impactful study.*

Thank you for your suggestions. We have incorporated the answers to these general comments and questions into an added Discussion section which we feel strengthens the impact and clarity of the study. We have addressed each point you have raised separately below and linked it back to text we have added or changed from the revised manuscript.

**NCEP vs HadGEM2-ES behaviour**: We have included a section showing the differences in the NCEP and HadGEM2-ES forcing fields and have tried to come back to this in the results and the discussion sections. As would be expected there are differences in the atmospheric fields,

from Section 2.2, second paragraph:

"The surface air temperatures are significantly higher in the NCEP reanalysis than HadGEM2-ES between December to early April in both the 1980s and 2010s period...

The shortwave and longwave radiation values are very different between the two data sets. NCEP has much higher summer shortwave radiation values, whilst HadGEM2-ES has much higher year-round longwave radiation values, but particularly summer values. It is likely that this dramatic difference is due to differences in cloud cover."

Then later we link the differences in melt rates to the difference in the surface air temperature field in Section 3.2:

"Melting occurs first in the outer regions (*always MIZ* regions) and progresses inwards across the sea ice cover to the *always pack ice* region. This is more pronounced in the NCEP simulation, likely a reflection that the NCEP atmospheric forcing is warmer in summer (see Figure 1a)."

"Peak melting rates increase, particularly in the NCEP simulation where they increase by 49%, compared to a 17% increase in the peak melting rate in the HadGEM2-ES simulation. The difference is likely due to the greater summer surface air temperatures (see Figure 1a) which drives a larger seasonal sea ice cycle (see Figure 4a)."

We then come back to discuss the differences between the NCEP and HadGEM2-ES forced simulations in the added discussion section:

"We chose to run simulations with the NCEP and HadGEM2-ES forcing so that the NCEP forced simulation could act as a check on the HadGEM2-ES forced simulation, which is projected to 2050. The two simulations were relatively similar in terms of sea ice extent, MIZ extent (see Section 2.4) and MIZ fraction (see Figure 7). Largely the overall results and proportions of growth and melt were similar in the two simulations, however the changes between the 1980s to the 2010s were generally larger in the NCEP forced simulation. This includes the changes in volume fluxes in both regions (see Figures 8a-f), reflecting the larger reduction in summer sea ice volume between the two periods (see Figure 5a). The differences between the NCEP and HadGEM2-ES forced simulations volume changes are largely a reflection of HadGEM2-ES having a much lower sea ice volume in the 1980s, however as the simulations become closer over time, it is difficult to assess whether the HadGEM2-ES simulation is underestimating the change from the 2010s to the 2040s."

**MIZ vs non-MIZ mass budgets and reasons for the differences**: We have tried to incorporate the answer to this in several places throughout the manuscript. Starting with in the introduction we have included paragraphs detailing MIZ processes and a qualitative description of contrasts we might expect to see based on the definition and conditions in the MIZ:

"The strength of sea ice is strongly dependent on the SIC. For 80% SIC (the upper MIZ boundary), we can estimate (Hibler 1979) that ice strength is less than 2% of its

maximum. In the MIZ internal stresses in the ice play only a small role and sea ice is essentially in free drift. The sea ice in the MIZ behaves distinctly to pack ice as it can be more easily advected. This has implications for those wanting to cross the Arctic: a larger Arctic MIZ would be easier to send ships across.

The larger concentration of smaller floes and lower sea ice concentration in the MIZ has a number of consequences for the sea ice interactions with the ocean and atmosphere. Lateral melting will be enhanced due to the increased perimeter to surface area ratio, creating open water more efficiently than top or basal melt, and potentially fuelling the ice-albedo feedback. The lower the ice concentration, the more the surface ocean is warmed due to the lower albedo of open ocean, further enhancing ice melt and leading to the positive ice-albedo feedback. The increased open water fraction can also mean an increase in wind mixing in the mixed layer and will affect Arctic Ocean spin up (e.g. (Martin et al 2016)). There is a wave-floe size feedback that means the smaller the floes, the larger the impact of the waves, so a positive feedback loop exists that can act to increase the action of waves on the sea ice floes and further increase the concentration of smaller floes. The location and volume of sea ice melt has implications for stratification and so how deeply solar heat is mixed down. More sea ice melt means the mixed layer is shoaled and solar heat is concentrated in the upper water column.

Meanwhile there are other important sea ice processes, such as top melting where it is less clear that we would expect there to be a contrast between the MIZ and the pack ice, for example in the formation of melt ponds. In the Arctic, the snow thickness is generally modest compared to that on Antarctic sea ice, and the location of top melting and the formation of surface melt ponds is primarily driven by atmospheric conditions. Projections suggest that the MIZ will increasingly dominate the Arctic sea ice cover, especially in summer. It seems likely, therefore, that MIZ-focussed processes will play an increasing role in controlling the mass budget of Arctic sea ice."

We have then revisited these points in the Discussion section:

"The CICE model set up we used is relatively physics rich, which we believe is needed to represent the contrast between the pack ice and MIZ, as well as some of the changes over time. The differences in lateral melting was very likely caused by the inclusion of the FSTD model (Roach et al., 2018, 2019; Bateson et al., 2022), although we did not directly test this within this study. It is possible that lateral melt might be increased by the inclusion of a full wave model, though Bateson et al. (2022) show brittle fracture is likely just as important, and more so in the pack ice. The increase in top melting in the 2040s in the projection supports the importance of the topological melt pond scheme (Flocco et al., 2010, 2012) that we use, and the increasing role they play in the sea ice mass balance and evolution. As an increasing fraction of the summer sea ice cover becomes MIZ we expect that the

representation of FSD-wave interactions and the melt processes themselves is likely crucial to realistically representing Arctic sea ice and the transition to sea ice free summers. The representation, or lack of representation, of such processes can contribute to discrepancies of Arctic sea ice (Diamond et al., 2021)."

**Approach of splitting the ice cover into MIZ and non-MIZ regions and new insights from this:** The comments about processes in the MIZ in the introduction included in the previous comment help to introduce why we would want to use this approach why we might expect differences between the regions through the description of sea ice processes in the MIZ. We have then added a paragraph in the discussion to add some of our thoughts on the use of the method:

"Our results show that sea ice processes do have a dependence on ice concentration, as would be expected, supporting the separation of the sea ice cover into MIZ and non-MIZ regions for analysis of the volume budget. Our results also indicate that if we separated the MIZ (and the pack ice) into more ice concentration based categories we would see distinct behaviour in the balances of processes, particularly in the type of melting. The MIZ and pack ice divide we have used differentiates between where internal stresses becomes important (SIC>80%), and where they become small in the MIZ, and the sea ice is in free drift. Our approach also has the advantage of simplicity, the more categories the MIZ is split into, the more complex the analysis becomes and the less clear the results. We believe we have struck a balance between the complexity required and keeping the analysis as simple as possible to understand. Although the ice is more dynamic in the MIZ, it was shown in this study to be a decreasing sea ice sink term due to the reduction in sea ice volume, meaning there is less sea ice to transport. Additionally. there was a relative increase in the melt terms, see Figure 8."

**Insights on broader questions:** We have added some comments on the importance of the inclusion and development of processes that are more dominant in the MIZ to the Discussion and Conclusion. Lines added to the discussion:

"The CICE model set up we used is relatively physics rich, which we believe is needed to represent the contrast between the pack ice and MIZ, as well as some of the changes over time. The differences in lateral melting was very likely caused by the inclusion of the FSTD model (Roach et al., 2018, 2019; Bateson et al., 2022), although we did not directly test this within this study... As an increasing fraction of the summer sea ice cover becomes MIZ we expect that the representation of FSD-wave interactions and the melt processes themselves is likely crucial to realistically representing Arctic sea ice and the transition to sea ice free summers. The representation, or lack of representation, of such processes can contribute to discrepancies of Arctic sea ice (Diamond et al., 2021)."

Lines added to the end of the conclusion: "Our analysis demonstrates a different balance of processes control the volume budget of the MIZ versus the pack ice. They

are understandable in terms of the physical processes that dependent on the ice concentration, such as wave-ice interaction and lateral melt, which we are able to account for in our relatively physics rich sea ice model. We suggest that representation of such processes, in models such as climate models, requires more attention as a greater fraction of the sea ice cover becomes MIZ."

*Specific comments.*

1) *Please provide more information throughout the manuscript on the value of separating analysis into MIZ/non-MIZ regions. How do you expect processes to differ in these regions? Why do things differ across these regions both in their mean state and in their response over time? What value is added by looking at the system from this perspective?* A section has been added to introduction discussing processes that might be expected to differ between the MIZ and pack ice, see answer to General comments, **MIZ vs non-MIZ processes** section and **Approach to splitting the sea ice cover in MIZ and non-MIZ regions**. We have also added a Section on the atmospheric forcing used, to show the warming being applied, and a Discussion where we come back to the approach we have used.

2) *Please provide more information on what new information/insights were learned from this study relative to previous work.* Whilst the Keen et al study looked at mass budget between different climate models in the mean state and over time, we carried out the first study to our knowledge studying the MIZ/non-MIZ. We believe this is important to studies interested in the MIZ itself and those studying the ice cover over periods when the MIZ is a dominant part of the summer sea ice cover. In particular, when thinking about which model use to study this period. We believe it also highlights future work which could be done to investigate the feedbacks between waves-FSD-lateral melting further in the MIZ. We have discussed some of these points in a discussion, see the manuscript excerpt in the response to the General comments, **Insights on broader questions** section.

3) *Please provide more information on the implications of these new findings for bigger questions on the future evolution of the ice cover and/or discrepancies across models (for example).* I have added a Discussion section and some discussion of the importance of representing different processes, see answer to General comments on **insights on broader questions**.

4) *Line 25, "if" should be "is"* Corrected, see line 25.

5) *Lines 52-53: I appreciate that you acknowledge the limitations of the forced model framework. I think that it would be useful to return to this in the results or conclusions section to discuss how these limitations may specifically affect the results from this study.* We have added this to a discussion to revisit the implications. Lines 285-326:

"Our forced sea ice-mixed layer model receives no trend in subsurface ocean properties, such as the "Atlantification" of the Arctic as the subsurface Atlantic Water layer becomes warmer and thicker (Grabon et al., 2021), which has the potential to cause sea ice loss if the heat reaches the surface (Polyakov et al., 2013; Onarheim et al., 2014; Carmack et al., 2015). It is possible that some of the relative increase in top melting could be due to the constant ocean forcing, which may lack some ocean warming that we might expect to see in the 2040s. Although how much of this is mixed into the upper layer that interacts with the sea ice is an open question. Additionally, field observations indicate that the majority of the ocean heat needed to explain basal ice melt rates can be explained from solar radiation (Perovich et al., 2011), something our model does capture. Note that using coupled and climate models introduces its own set of problems, e.g. CMIP6 models fail to simulate a realistic seasonal cycle of sea ice area (Notz et al., 2020). Using a forced sea ice model allows us to simulate a more realistic sea ice state, which has been shown to affect the balance of sea ice processes (Holland et al., 2010; Keen et al., 2021)."

6) *Line 90-92: "The ocean temperature and salinity below the mixed layer …" Does this lead to a weaker sea ice response, especially near the ice edge? How do the results from this study compare to the sea ice simulated in the HadGEM2-ES runs?* The restoring is required to account for horizontal ocean heat transport. A 20-day time scale does not affect the ocean-ice interactions on shorter time scales. The sea ice response is not weaker, it is the mixed layer properties that impact the sea ice. This modelling approach for the mixed layer has been used in several other studies, see lines 87-89 in the revised manuscript. In the original coupled HadGEM2-ES simulation, the MIZ is typically smaller, also not in the right places, as it is simulated all the way around the ice cover at a relatively constant width, including at all coastlines. The difference in coastline and sea ice model, means that comparing our simulation to the original HadGEM2-ES simulation does not show us the impact of using a fixed T/S below the mixed layer on ice loss. We have included a Figure below showing how sea ice and MIZ extent compare in the summer months, using only the regions that CICE also simulates sea ice. The HadGEM2-ES simulation (orange line, thin lines are sie and thicker and MIZ extent) does have similar sea ice extent in August and September but is lower than our simulation (red line) in June and July. The MIZ extent is much higher than observations and the simulations in June (for the reasons listed above), but is more similar for the other months, although the MIZ being compared is not in the same places.

[Figure]

7)  Line 99-100: "and a prognostic floe size distribution model" What is the wave forcing used to drive this model? Are there any feedbacks from the model onto the wave forcing? I assume not since this coupling is not mentioned. The discussion of the limitations of this, particularly for studying things in an MIZ/non-MIZ perspective should be discussed.

We have now added some description of the wave model and wave forcing used, see lines 96-105:

"We used the same wave forcing set up as Bateson et al., (2022), note that this is a different set up to Roach et al., (2019) where a separate wave model is coupled to the sea ice model to calculate the wave properties in the grid cells that contain sea ice. Instead, we use an extrapolation method as used and documented in (Roach et al., 2018), where ERA-I wave forcing (Dee et al., 2011) is used to calculate the necessary in-ice wave properties. The wave forcing consists of the significant wave height and peak wave period for the ocean surface waves. These fields are updated every 6 hours in the grid cells that contain less than 1% sea ice. Crucially for this study, despite not having a coupled wave model our set up still enables wave induced fracture causing enhanced lateral melting and wave-dependent new ice formation, as outlined in Roach et al., (2019). After 2017 we repeat the wave forcing, which does mean there is no trend in the wave forcing. Sensitivity studies varying the

wave forcing using this model have demonstrated limited sensitivity to the wave forcing (Bateson. 2021) and comparisons to future 2056-2060 climatology from a global RCP8.5 wave simulation shows no significant change in significant wave height or interannual variability in significant wave height (Bateson. 2021). Although the wave forcing fields do not have any trends, the propagation of the waves into the ice field does respond to the changes in the ice cover over time. The simulations were initialised with a 6 year spin up period, this is a similar length to previous studies using the same model setup (Rolph et al., 2020; Bateson et al., 2022). As we are using a standalone sea ice model, the amount of spin-up required is much shorter than a climate simulation, or a coupled sea ice-ocean model."

8) *Line 102-103: "The HadGEM2-ES product ..." Could you say a bit more about this product? Is it from a single ensemble member? How does this forcing (for example, Arctic mean surface air temperature, precipitation, etc.) compare to observations?*

There were three members of the ensemble, we used the first one. We have now added a section on the forcing sets where we compare it to the NCEP reanalysis, see Section 2.2 in the amended manuscript. Keen and Blockley (2010) includes a short summary of a comparison of the main atmospheric fields and biases present, but it is considered to be reasonably realistic.

From Keen and Blockley 2010:

"HadGEM2-ES is a coupled atmosphere–ocean model that was submitted to CMIP5. The model includes interactive atmosphere and ocean carbon cycles, dynamic vegetation, and tropospheric chemistry (Martin et al., 2011; Collins et al., 2011). It is considered to have a good depiction of present day global cloud characteristics (Jiang et al., 2012) and the best model depiction of Arctic cloud and surface radiative forcing (English et al., 2015). The mean Arctic ice extent lies within 20 % of observed values at all times of the year, although the September extent has low bias and the magnitude of the seasonal cycle is too large, consistent with biases in winter net surface long wave (LW) and summer net surface short wave (SW) radiation (West et al., 2018)."

9) *Line 116-118: "The maximum sea ice extent ...": The winter ice edge (and maximum ice extent) are particularly sensitive to ocean conditions. How does the use of fixed T/S below the mixed layer influence ice loss to 2040? How do the results in this forced model framework differ from the HadGEM2-ES simulations and what does that tell you about the role of coupling? Similar questions are also relevant for the summer ice loss in the HadGEM2-ES forced runs and I'd suggest that you compare those also to the coupled HadGEM2-ES simulation that was used to obtain the atmospheric forcing data.*

See answer to point 6. It is difficult to make any conclusions from our study about the role of coupling on simulated future sea ice loss as there are differences in the sea ice model.

10) *Line 118-119. "The NCEP forced simulation shows a stronger declining summer sea ice extent trend …". Why is this the case? Is it consistent with internal variability? Is it due to biases in HadGEM2-ES forcing?*

The NCEP SATs are considerably warmer in Jan-April in all decades, and a larger warming between these periods in these months, which is likely the cause of the stronger trend in the NCEP forced simulation. See Figures 1 and 2, which have now been added to the manuscript.

11) *Table 1. For the 1980s and 2010s, it would be useful to also include observations on this table. Adding the range of values for the individual years in the 5-year periods that are analyzed would also be helpful for putting the differences between runs into context. It would also be helpful to show this information visually with a bar chart or something similar.*

We have removed this table and replaced it with a figure of the average annual cycle for each period of sea ice extent and MIZ extent. The new figure includes the observations and shows the differences between the study periods in each product/simulation. See Figure 4 in the revised manuscript.

12) *Figure 1. The quality of this figure should be improved. It is hard to distinguish the MIZ lines on the plots (particularly for Aug and Sept). What do the vertical bars on the observed extent signify?*

The vertical bars were error bars (+/-10% of sie), following the same method as Rolph et al.,( 2020). However, to make the plot clearer we have removed them and changed the sea ice extent line to half line thickness and the MIZ line to solid, which is hopefully clearer to read. See the revised Figure 3.

13) *Line 121. "NASA Team and NASA Bootstrap" – please provide a reference for these datasets.* References added, see lines 131 and 132 in Section 2.3.

14) *Lines 133-134. "By the 2010s the MIZ becomes the dominant part of …" Please clarify for what months this is true.* This is now clarified in the text, it is true for July, August and September. See line 159.

15) *Line 134-135. "the summer sea ice cover is almost entirely MIZ …" Please be more specific on what months this is for.* This has been clarified in the text, see lines 160-161.

16) *Line 138. "PIOMAS" – please provide a reference and brief information on the PIOMAS product. In particular, it should be noted that PIOMAS is a model product and not*

*direct observations.* We have now added a section on model validation data, see Section 2.3, where we have noted that PIOMAS is a model product.

17) *Line 165. "By the 2010s the MIZ" – please provide a comparison to the observations here.* The comparison to observations has been added in a new figure (replacement to Table 2 suggested by reviewer 2), see Figure 7.

18) *Line 175. "basal growth makes up the largest proportion of melting" - Do you mean "basal melt" here?* Thank you, this was a typo, changed to 'basal melt', see line 246.

19) *Line 177: "significantly more" – what is the significance level used to determine this?* "Significantly" was not being used in a statistical way here, we have changed it to "substantially" to avoid confusion, see line 221.

20) *Line 186. "This means that it is likely that MIZ closer to the pack ice has a different balance of processes to the outer MIZ that has lower ice concentrations." Given this, what is the value in separating things into a MIZ/non-MIZ framework? Is it useful? What are the limitations? Would it be beneficial to look at the mass budgets as a function of ice concentration instead?* We have now added some comments on this to a new Discussion section, see Section 4 and the response to General Comments, **Approach of splitting the ice cover into MIZ and non-MIZ regions** section.

21) *Line 192-194. "A possible explanation for the increase in top melting in the future MIZ…" Is this inconsistent with the results from Keen et al. that did include active ocean models? Are there other factors that could explain increased top melt? Do factors like the change in seasonality, regionality play a role? Could you provide more analysis (for example of heat budgets) to gain a better understanding of the factors at play?*

The increase in top melting is very likely driven by the increase in surface air temperature, see added Figures 1 and 2 in the new manuscript. The increase in top melting is consistent with the Keen et al paper, which also found an increase in top melting in the near future. We have now made this clearer in the text, see lines 227-231 and also in the Discussion section in Section 4.

22) *Line 194-197. "This means that the results here could be seen as a lower estimate on the ice loss…" I would encourage you to also assess the HadGEM2-ES runs to determine if those exhibit different ice loss characteristics. I appreciate that the ice models are different in these runs which will affect the results but it would nonetheless provide a useful point of comparison on the sea ice response with coupled ocean feedbacks.*

See response to point 6 and attached figure.

23) *Line 198. "convection and ridging". Isn't it just advection of the ice that can cause a mass flux? Won't ridging affect the distribution of ice but conserve ice volume/mass?* Yes, this is correct we have amended this in the manuscript to just advection.

24) *Line 229-230. "Peak total melting rates …" Why are there these differences between the NCEP and HadGEM2-ES runs?*

We have added some comments on differences between the simulations in the Discussion section, linking to the differences in the atmospheric forcing fields.

25) *Line 234-236. "In the NCEP simulation there is a 17% increase, whilst in the HadGEM2-ES simulation there is a 6% decrease, likely due to the larger melt rates in the NCEP simulation." Why "due to the larger melt rates"? How do the differences in melt rates drive the different ice growth response?*

We have rewritten this to remove the implied causality and added some NCEP-HadGEM2-ES comparison analysis in the discussion, see **NCEP vs HadGEM behaviour** in the answer to the General comments.

26) *Line 268-270. "Top melt was twice as important …" Please provide information on why these differences are present.*

It is likely these differences are driven by a combination of the differing FSTD in the MIZ vs pack ice, and the variation of warming over time spatially. The regions where pack ice is located have warmed more, driving more top melting. Meanwhile the presence of more smaller floes drives more lateral melting in the MIZ. We discuss some of these processes now in the Introduction, see lines 32-49 and also in the added Discussion section.

27) *Conclusions section. Please provide information on what new insights were gained in this study relative to previous work and the broader implications of this study.*

We have revised the Conclusion section and that has included making some comments on broader insights: "Our analysis demonstrates a different balance of processes control the volume budget of the MIZ versus the pack ice. They are understandable in terms of the physical processes that dependent on the ice concentration, such as wave-ice interaction and lateral melt, which we are able to account for in our relatively physics rich sea ice model. We suggest that representation of such processes, in models such as climate models, requires more attention as a greater fraction of the sea ice cover becomes MIZ."

---

## Referee Report (RR1)

**Toward a marginal Arctic Sea ice cover: changes to freezing, melting and dynamics**

Since its last iteration, significant changes have been made to improve the clarity of the manuscript, the quality of the figure, and the presentation of the result. For that, I congratulate the authors for the effort that they have placed into assessing the reviewer's comments.

However, the clarification of the results has raised concerns regarding the scientific content that were not apparent in my initial review. Those issues must be addressed by the authors by doing additional analysis and it should be either acknowledged or clarified within the manuscript.

Additionally, I believe that the manuscript still needs improvements regarding the writing and the structure. While the changes proposed in the review are probably not all necessary, I believe they would enhance the effectiveness of the message and the quality of the scientific communication.

I maintain my opinion that the reviewed paper deals with the scientifically important research question of the widening of the marginal ice zone and the transition towards the ice-free Arctic. However, I believe that major revisions are still required to reinforce the analysis and improve the text.

**Scientific concerns**

1) I was expecting larger differences between the volume fluxes in the "always pack-ice", "become MIZ" and "always MIZ" regions, especially for the partitioning between top, basal and lateral melt (Figs. 8-10). It is mentioned in the text that "the partitioning of the melt between top, basal and lateral melting does differ substantially between the pack ice and MIZ", but I must admit that I was not convinced about that by looking at the figures. As far as I am concerned, the changes in the volume flux (Fig. 8) and fraction of volume flux (Fig. 9) between the "always pack" and the "always MIZ" region (e.g. Fig. 8a and Fig. 8c) are almost of the same order of magnitude as the changes in volume flux simulated due to variations in the atmospheric forcing (e.g. Fig. 8c and Fig. 8f).

    I suspect that it might be related to the use of fixed regions for the analysis that are only based on the July SIC (Fig. 6). I believe that larger, and more significant changes could be shown by computing volume flux in regions that evolve monthly. While I don't think that doing such an analysis would affect the main message of the study, I believe that it would show more contrasting and interesting results.

2) In my experience, the CICE's FSTD model has shown to be extremely sensitive to the wave forcing used. The realism of the simulated mean floe diameter (MFD) pattern can often be questionable (e.g. Roach et al. 2018, 2019), and FSTD has been hypothesized to be biased toward generating small floe size. This unresolved issue is likely related, among other factors, to an unphysical parametrization of the wave fracture.

While this is not in the scope of the study, I believe it's important for the authors to acknowledge that the wave forcing has a major impact on the FSTD and, consequently, potential implications for the results, given that lateral melt depends on the MFD. I find the approach to handle the wave forcing in the study (i.e. prescribing a cyclic forcing from ERA-I) questionable.

First, the authors mention that "although the wave forcing fields do not have any trends, the propagation of the waves into the ice field does respond to the changes in the ice cover over time". However, if the wave forcing is prescribed at each grid point from a climatology, how does it consider the change in the ice cover over time? Is the wave forcing only prescribed at the ice edge with an attenuation computed inside the ice cover? Then, how does it deal with a grid point that was in the pack ice in the 2010s, but that is ice-free in the 2040s? What wave properties are prescribed at that grid point then?

Second, while the wave forcing is cyclic, the atmospheric forcing (including the winds) is not. If so, it does mean that after 2017 the wind pattern might not match with the wave pattern even if, in reality, those two concepts should be tightly interconnected.

While those two concerns suggest that the wave forcing used in the HadGEM2-ES projection is unrealistic, it might not be significant for the results as long as the MFD remains somewhat realistic. Therefore, for the sake of the discussion, it would be beneficial to look at your simulated annual cycle of the FSDRAD field (mean floe diameter) in the 1980s, 2010s, and 2040s.

**General comments**

- While short sentences are typically preferred in a scientific document, I believe that the author might have pushed the limit a bit far in some sections. Reading the manuscript often feels a little bit robotic. I suggest trying to combine some of the short sentences to introduce variation in the sentence length throughout the manuscript and to enhance the rhythm of the paragraphs.

- The organization of the document is essential; the readers want to know where to quickly find the information they are looking for. Currently, results are spread between the methodology (section 2) and the results (section 3), which makes the story hard to follow.

  The methodology should be strictly about describing the data and the method used, so the reader has enough information to reproduce the results. This is not where you typically analyze data, even if it is for model validation. Therefore, I recommend creating a new result section called "Model validation" and moving Figs. 1,2,3,4,5,7, as well as their associated discussion to this new result section. Here is a suggestion for a preferred document organization:

1. Introduction

2. Data and methods: this should include what is currently sections 2.1, 2.2, 2.3 and 2.6.

   2.1.     Model set-up and forcings or Model configurations.

   *Include the content of what is currently sections 2.1 and 2.2.*

   2.2.     Model validation data or Observational data.

   *Include the content of the current sections 2.3.*

   2.3.     Analysis method or methodology

   *Include the content of the current section 2.6. Move lines 200-206 to the results.*

3. Results: this should include what is currently sections 2.4, 2.5, 3.1 and 3.2. The author should also consider having shorter and more evocative subsection title.

   3.1.     Model validation*.*

   *Include what is currently sections 2.4, 2.5 and the part of 2.6 mentioned previously and the discussion about Figs. 3,4,5,7.*

   3.2.     Annual total volume fluxes.

   3.3.     Annual cycle of the main melt and growth terms.

4. Discussion

5. Conclusion

- The length of the manuscript also increased a lot (i.e. it went from 15 pages to 21 pages). I believe that there are some lengths and redundancies, which can undermine the message. In the specific comments, I go through each of the sections of the manuscript, and I suggest changes and rephrasing to improve the conciseness and clarity.

**Specific comments**

**1) Abstract**

The abstract is the first part that most of the readers read and sometimes the only part that they will read. For this reason, it is probably the part of the manuscript that needs to be written with the utmost care. It needs to tell the story on its own while being as dense, concise, and as clear as possible.

- Line 1 and 4-5: Repetitive sentence; the authors should remove one.

- Line 5: Typo ? pack ice.

- Line 7: "*This is the first study (to our knowledge) that separately considers the pack ice and MIZ in this way.*" This type of sentence should be avoided; you either know that this is the first study, or you don't. The authors should conduct the appropriate literature review to make sure that it is or remove the sentence.

- Lines 5-10: the method of the study is summarized, however, it is not mentioned that two atmospheric forcing (NCEP/HadGEM2-ES) are used. Then, in lines 11-13 when the discussion of the results starts with a comparison between these two atmospheric forcings. In short, the part of the abstract that talks about the method should directly lead to the results.

- Line 9: "*The model has been compared to floe size distribution observations.*" Is this true? I have been reading the manuscript repeatedly and I do not see any comparison with floe size distribution observations. I only see comparisons with SIC and ice thickness observations.

- Line 10: The abstract usually only has one paragraph.

- The last part of the abstract lists some of the results, but it should be instead focused on the principal conclusion of the study (i.e. the take-home message).

I believe that the reorganization proposed in the general comment will help to really highlight the story of the paper. Here is an example of how that structure could be used

the convey the message more clearly in the abstract.

*"The marginal ice zone (MIZ), defined as the region of the ice cover that is influenced by waves, is projected to become a larger percentage of the summer ice cover as the Arctic transitions to ice-free summers. Here, we compare individual processes of ice volume balance in the pack ice to those in the MIZ to establish and contrast their relative importance and examine how these processes change as the summer MIZ fraction increases over time. We use CICE, a physics-rich sea ice-mixed layer model forced with two atmospheric datasets; the HadGEM2-ES simulation (1980-2050), to simulate the ice cover in a high emission global warming scenario and the NCEP reanalysis (1979-2020), for comparison during the observational period. First, we compare both simulations to satellite-derived sea ice concentration (i.e. NASA Team/Bootstrap) and PIOMAS estimates of sea-ice thickness. Results show that [results from section 3.1] Then, we compare the annual volume fluxes for the following periods: low MIZ (1980s), high MIZ (2010s), and all MIZ (2040s), showing that [results from section 3.2]. Finally, we look at the annual cycle of the main melt and growth terms [results from section 3.3]. Those results highlight that [the take-home message]."*

**2) Introduction**

The introduction does a decent job of stating the background information and covering existing literature, but I still believe that it could use some rewriting to enhance the quality of the scientific communication. Here is what I suggest the authors to keep in mind to improve the flow of the introduction.

First, "contextualize the background information"; it needs to go from the most general (i.e. general statement about sea-ice, the MIZ, global warming, etc.) to the most precise (i.e. things that are directly related to the paper, e.g. the processes affecting the sea-ice volume budget). Second, "state the problem"; this is where similar previous literature is cited, and the knowledge gap or general misunderstanding of the problem is stated. Third, "address the problem"; this is where the methods, the scope of the paper, and what is unique about it are mentioned.

Additionally, some statements are made without citation; the introduction could use additional references especially between lines 35-51.

- Line 23: applied is repeated twice. The authors should reformulate.

- Line 26: *"More fragile"* feels vague. Thinner, more mobile, or more fractured could be more accurate.

- Line 30: This is still the "contextualize the background information" part. I suggest moving the sentence to the "address the problem" part.

- Lines 32-35: This is a general statement about sea ice. I suggest moving this part to the beginning where the different SIC threshold for the MIZ definition (i.e. lines 22-25) is discussed.

- Lines 36-40: The word *"potentially"* should be avoided. This has been the subject of many studies. Add references.

- Lines 38-40: the use of *"ice-albedo feedback"* is repetitive. The sentences could be merged or rephrased to avoid repetition.

- Lines 40-45: again, the use of *"feedback"* is repetitive and vague as it is not clearly explained here. Also, the word *"action"* feels vague. Wave-induced fracture could be more accurate. See "Floes, the marginal ice zone and coupled wave-sea-ice feedback" (Horvat, 2022), to improve that paragraph. Add references.

- Line 49: Add references.

- Lines 52-64: This is the "state the problem" section of the introduction. While other papers that made a sea-ice budget are cited, more emphasis should be placed on the lack of knowledge that persists after those studies. Instead, here, some of their results are stated, but they do not appear to be directly related to the content of this manuscript. For example, "*They also found that the initial sea ice state was important in determining projected changes to the sea ice cover, with thicker initial ice resulting in more sea ice volume change.*" As far as I'm concerned, the manuscript does not investigate of the effect the initial sea-ice state on the volume budget, therefore the emphasize should not be placed on that. It should instead focus on things that are relevant to the findings of this study (e.g. do they compare their results to observations? Do they analyze the processes in the MIZ and in the pack ice separately? etc.)

- Line 65: This is the "addresses the problem" part of the introduction. Again, I suggest changing the phrasing "*the first to our knowledge*". That could be implied by improving the previous paragraphs.

- Line 69-70: The emphasis should be placed on the fact that CICE has been used in previous modelling studies that focused on the representation of the MIZ.

- Line 71-77: A grocery list to describe the content of the manuscript should be avoided. The author should instead make a story out of it or remove it. This especially applies to the last part which simply states that section 4 is called discussion and section 5 is called conclusion. I think believe that the structural change proposed in the general comment will help to make a story out of the results. Moreover, a similar grocery list is made at the end of the methodology (lines 207-209), I suggest keeping only one.

**3) Methodology**

As I mentioned before, the methodology should not contain any figures. The only figure that could potentially stay in this section is Fig. 6, as it serves more as a mean of better understanding the methodology rather than a result.

I recommend moving Figs. 1 & 2 to the appendix as well as the discussion of those figures, to reduce the length of the manuscript.

Moreover, the section is to be called "Data and methods", as this is also where the datasets used in this study are described.

- Line 81: what is CPOM.

- Lines 86-88: "*found to give realistic simulations of observed floe size distributions (FSD) for mid-range floe sizes in the Arctic. This model, minus the brittle fracture addition to the FSTD model, has been used previously by Rolph et al. (2020) to compare changes in the MIZ in a sea ice model to satellite observations.*" This should be mentioned in the introduction, especially where it is argued that CICE is a good model to represent the MIZ (i.e. lines 68-10). The method should instead only focus on describing the component of the model used.

- Lines 86-88: Also, I would be careful about saying that Roach et al. 2018,2019 give realistic simulations of the observed FSD as I do not think that there are any comparisons to observations in those studies.

- Line 90: There is no need to mention the full name of the ocean reanalysis, it can be found in the reference (e.g. "a *climatological ocean reanalysis (Ferry et al. 2011)*").

- Lines 91-92: A separate sentence for the currents is unnecessary, it can be merged.

- Lines 93-98: This paragraph can be further simplified especially if the namelist is given in the supplementary material or in the code availability section. For example:

  "*We use a number of the default CICE settings, including the layers thermodynamics of Bitz and Lipscomb (1999), Maykut and Untersteiner (1971) conductivity, Rothrock (1975) ridging scheme, the delta-Eddington radiation scheme (Briegleb and Light, 2007), and the linear remapping ice thickness distribution (ITD) approximation (Lipscomb and Hunke, 2004). Additionally, we use a prognostic melt pond model (Flocco et al., 2010, 2012) and an anisotropic plastic rheology (Heorton et al., 2018; Tsamados et al.,2014; Wilchinsky and Feltham, 2006).*"

- Lines 98-114: This deserves a new paragraph; this section might need to be reworked according to the answer to the second scientific concern comment. Additionally, the authors should reorganize the paragraph by first describing the way that the wave forcing is dealt with in their own study. Then, they can briefly mention how others have done this differently and why they argue that their method is adequate. For example, the paragraph should start with:

  "*The wave forcing used in this study is prescribed from on ERA-I reanalysis wave data and is repeated after 2017 (Bateson et al. 2022). The wave properties used are the significant wave height and the peak wave period, which are then extrapolated and updated every 6 hours in grid cells that contains less than 1% sea ice (Roach et al. 2018). This wave forcing set-up then differs from Roach et al. (2019) […]*"

- Lines 106-109: I am not entirely convinced that the absence of trend reported in Bateson (2021) is a consensus. Many studies argue that the increase in open water in the summer that creates a larger fetch, combined with an increase in the intensity and frequency of storms in high latitudes will lead to waves of larger amplitude in the ice-free Arctic. The authors should read the following papers: *Swell and sea in the Emerging Arctic Ocean* (Thompson et al. 2014), *Growth of Wave height with Retreating Ice Cover in the Arctic* (Li et al. 2019), *Sea Ice Retreat Contributes to Projected Increases in Extreme Arctic Ocean Surface Waves* (Casas-Prat et al. 2020) and *Wind and wave climate in the Arctic Ocean as observed by altimeters* (Liu et al. 2016). It should be at least acknowledged in the paragraph that there is no consensus on that yet.

- Lines 102-103: "*The wave forcing consists of the significant wave height and peak wave period for the ocean surface waves*". If the same FSTD model as in

Roach et al. 2019, the CICE fracture model uses a wave spectrum to compute fracture, how is the wave spectrum derived from significant wave height and peak wave period only?

- Line 110: I suggest making a new paragraph for the discussion of the spin-up (i.e. lines 110-113) or moving this section after the discussion on the component of the model (line 99).

- Line 115: At this point, the author should have already made clear that they are comparing the result from two atmospheric forcing before as this is a key part of the story. It needs to be mentioned in the abstract and the "address the problem" part of the introduction.

- Line 136: Add references.

- Figs. 1 & 2 are complementary to the main findings, but they are not essential for the story. I suggest moving them to an appendix with lines 124-136. The figure can be referred to later, when NCEP and HadGEM simulations are compared in the results. Lines 115-123 can be moved to the "*Model configuration*" section.

- Fig. 1: The colors are confusing. The author should consider using a dashed line NCEP and a solid line for HadGEM (or vice-versa).

- Line 138: If Rolph et al. (2020) have already done that comparison, why doing it again? The author needs to clearly highlight what is new about their paper and it must be clear from the "state the problem" part of the introduction.

- Lines 139-145: A basic description of the two SIC datasets used for validation should be there. How exactly is it measured? Is it both passive microwaves? Which satellite? Why is there so much difference between the datasets?

- Line 145: *"Daily values are used to compute monthly values of sea ice and MIZ extent that are plotted in Figure 3."* This should be in the caption of the figure instead.

- Fig. 3: I am not sure of the necessity of: *"Uncertainty levels of +/−10% were used for the satellite values in Rolph et al. (2020), they have been left out in these figures to make them clearer."* as it is not discussed in the text nor shown in the figure.

- Lines 146-151: Most of it can be cut as it is general sea-ice knowledge. The paragraph could be simplified as:
  *"To validate our simulated sea-ice thickness, we use the Pan-Arctic Ice Ocean Modeling and Assimilation System (PIOMAS) (Zhang and Rothrock, 2003), a model that assimilates a range of sea ice area/concentration observations to give an estimate of continuous Arctic Sea ice volume changes over time. As with the satellite data, PIOMAS has been interpolated on the ORCA tripolar 1∘ grid and the CICE land mask has been applied."*

- Lines 153-155: Most of it is already in the caption of Figure 3, it can be deleted and rephrased.

- Line 158: The sentence *"slighter weaker sea-ice trend"* is unclear. Is it really about the trend (i.e. the rate at which the sea-ice extent changes), or about the sea-ice extent as shown in the figure? Also, is it about the MIZ extent or the total sea-ice extent? The author can be more precise when referring to a figure (e.g. *"slightly weaker MIZ extent trend (see thick line in Fig.3)"*).

- Lines 158-159: *"Whilst"* is used in two subsequent sentences. Rephrase.

- Line 164: *"An increasing trend"* in what? Again, the authors can be more precise in their statement.

- Lines 178-179: Unnecessary, this is already mentioned in the caption. Also, this part does not require a separate section, just add that paragraph to the previous section and call it "model validation".

- Lines 187-189: Again, this is unnecessary as it is already mentioned in the caption of the figure. This comment applies to most of the figures. For example, the paragraph could be simplified as:
  *"We consider three different ice cover states within the simulations: a low MIZ state in the 1980s; a high MIZ state in the 2010s; and an all MIZ state in the 2040s. In each case we use the last 5 years of daily July SIC and assign each grid cell as pack ice (SIC≥80%), MIZ (15% ≤SIC<80%) or open water (SIC<15%) (Fig. 6).*

  The authors should avoid breaking the flow of the paragraph with sentences like *"Figure 6 shows … [content of the caption of Fig. 6]."*. This is repetitive as the reader will already look at the caption to understand what is in the figure.

- Line 184: Typo ? *"for the scope of this study."*

- Line 198: This is already mentioned in line 141.

- Lines 200-206: This should also be moved to the new "model validation" results section as well as Fig. 7.

- The caption of Figure 6 can be simplified. Change *"Region 1 (blue) indicates the area that is pack ice in both the 1980s(2010s) and 2010s(2040s) in subplots a-d(e). Region 2 (green) indicates the area that is pack ice in the 1980s(2010s) and becomes MIZ in the 2010s(2040s) in subplots a-d(e). Region 3 (orange) indicates the area that is MIZ in both the 1980s(2010s) and the 2010(2040s) in subplots a-d(e)."* to *"Region 1 (blue) indicates the area that is pack ice, region 2 (green) indicates the area that is pack ice and becomes MIZ and region 3 (orange) indicates the area that is MIZ".* Identify the period in the title of the subplot (e.g. (a) Bootstrap 1980s-2010; (b) NASA Team 1980s-2010; (c)… etc.)

- Lines 207-208: Unnecessary, it could be removed.

**4) Results**

The figures and the description of results are much clearer than in the last iteration and the results paragraphs are straightforward. However, as I mentioned previously, the author should avoid repeating the content of the caption in the text when referring to a figure.

Also, when referring to a figure, direct and indirect citations are possible. The direct method is to directly describe the figure in the text. For example.

"The annual volume fluxes for sea ice processes are shown in Figure 8.",

The indirect method is the cite the figure in parentheses at the end of the sentence that describes the results.

"Analysis annual volume fluxes for sea ice processes show that congelation growth dominates sea ice growth […] (Fig. 8)."

More indirect references should be made throughout the text, especially for figures that do not require a lot of explanation, as it is less wordy. This will also help to avoid repeating the content of the caption. Also, while using the indirect approach use the abbreviation Fig. instead of Figure to lighten the text.

Finally, the author should consider the opportunity of using the indirect approach to add additional and more detailed references to the figures throughout the result section.

The description of the figures can often be confusing, especially for Figs. 8-10 which contains many panels, with many different lines and bars.

- Fig. 8: It is unnecessary to repeat the *"Volume flux"* in the y-label three times in the subplot. I suggest putting one y-label for the whole subplot.

- Fig. 9: "Fraction of the melt" in the y-label is inaccurate as the dynamic term is not melting. "Fraction of the total ice loss" would be more accurate. Again, the y-label does not need to be repeated. In the caption, I suggest the following phrasing: "*The evaporation term is neglected.*"

- Line 257: "*The same regions and time periods defined in Figure 6 have been used.*" Already mentioned in the caption.

- Line 270: The threshold for the melting should be mentioned at the beginning of the paragraph. Is this threshold applied to the total melt or to one of the individual melt terms?

- Fig. 10: I believe that the y-label is incorrect. In Figure 8, the y-label is volume flux with units of m3/m2 and then in Figure 10 the y-label is also volume flux, but now with units of m3/m2 days. In the text, the author refers to growth and melt rates in lines 284-299 and I think that they are referring to Figure 10. I believe that the y-label should be growth/melting rate, but this needs to be clarified.

- Line 272: Which of the melting rates are you talking about? The total (black), the top (red), the lateral (purple) or the basal (yellow)?

- Lines 278-298: Are those results related to Figure 10? If so, refer to the figure in the text, at least one reference per paragraph. For example,
  "*In the always pack region peak melt increases and the melt season gets longer in the 2010s relative to the 1980s by 13 days in the NCEP and 6 days in the HadGEM2-ES forced simulations (Fig. 10a,d).*"

5) **Discussion and conclusion**

The material in the discussion is adequate. I believe that the two last paragraphs of the discussion could be improved by considering all the changes mentioned previously and by adding more direct references to the results or to previous literature (just like in the two first discussion paragraphs).

In the conclusion, the author is doing a great job at summarizing their results. However,

I believe that the authors are missing the opportunity to emphasize a clear take-home message. Which one of those results was unexpected, novel, or significant? Which unanswered questions were addressed in the manuscript? What are the key findings uncovered by the methodological framework? This ties back to my earlier comment in the introduction about finding the knowledge gap in the literature, which I believe is never clearly highlighted in the manuscript.

- Line 305: New paragraph?

- Line 323: *"The differences in lateral melting were very likely caused by the inclusion of the FSTD model (Roach et al., 2018, 2019; Bateson et al., 2022), although we did not directly test this within this study."* I believe that your methodological framework allows to investigate such a question for example, by comparing the mean floe diameter in the MIZ and pack ice regions. My concerns about the wave forcing should be addressed or acknowledged here.

- Line 325: New paragraph?

- Lines 331-338: While the other paragraphs bring interesting discussion points (e.g. the use of a coupled climate model vs a forced sea-ice model, the absence of warming trend in the oceanic forcing, the effects of using an FSTD and a melt pond scheme), I am struggling to understand what is the point that the author is trying to make in that paragraph. For example, with the sentence: *"Our approach also has the advantage of simplicity, the more concentration categories the MIZ is split into, the more complex the analysis becomes, and the less clear the results."* This appears to be a simple general statement that does not integrate or extend any of the results.

- Lines 343-344: Rephrase.

- Lines 344-345: *"The MIZ is defined as having a sea ice concentration (SIC) between 15% and 80% and pack ice is defined as SIC>80%."* I believe that this should be included in the first sentence of the conclusion.

- Line 370: Typo? "particularly in the region that is remains MIZ."

- Line 374: Typo? *"Our analysis demonstrates a different balance of processes control the volume budget…"*

- Line 374-378: This is a rather vague and general way of closing the paper. I believe that a stronger concluding remark could be made by including future

research directions.

---

## Author Response (AR2)

**Response to Referee Comments**

*Referee comments are shown in black, our response is in blue, and changes to the manuscript are shown in red.*

We thank the referees for their comments. We have considered all these comments carefully and have made many improvements to the scientific content and organisation of the manuscript. The volume of work required is such that we have altered the order of the authorship to reflect the additional work we have done.

**Referee 1**

Firstly, we would like to thank the referee for providing thorough and thoughtful comments on our manuscript. They have been invaluable in improving the manuscript.

I thank the authors for substantially revising the manuscript in response to the review comments. The structure of the paper is much improved and more useful context has been provided for the study motivation and results. I still believe that additional discussion is needed on why some mass budget changes occur, why this differs for the MIZ/non-MIZ regions, and what this implies for the value of looking at simulated mass budgets from the perspective presented here. Additionally, clarification is needed for certain statements made within the manuscript. I provide more specific information below. I believe that a publishable manuscript will be possible after these comments are addressed.

We appreciate the recognition of the improvement in quality of the manuscript since the original submission and are grateful to the referee for their role in this process. The remaining concerns raised above have been addressed in detail via our responses to the comments below.

**Specific comments**

Line 17. "decreases" should be "decrease"

Corrected as suggested.

Line 41. "spin up".

What do you mean here by "spin up"? I assume that you are referring to the possible changes in ocean circulation. Please clarify.

Yes, this is standard terminology in Arctic oceanography and refers to changes in the ocean circulation (e.g. Giles et al., 2012).

Line 92. "mixed layer temperature, salinity and depth".

Could you clarify what type of ocean mixed layer model is used for this?

We use the prognostic mixed-layer ocean model of Petty et al. (2014), which is stated in the first line of the previous paragraph. Mixed layer temperature, salinity, and depth are all prognostic parameters in this model.

We have added some additional clarifying details where we first reference the mixed layer model in this paragraph.

Line 118. "monthly averages are used for precipitation"

Why only use monthly averages? Could a lack of daily information be important for not adequately representing the snow to rain transition with implications for the surface albedo? How is the precipitation phase determined? I appreciate that this may be a limitation of the forcing data available but a short explanation here would be useful and provide some context for the potential impacts of this on the simulations.

There is a high uncertainty associated with precipitation in the reanalysis, however the uncertainty will be lower in a monthly average than in daily / hourly precipitation values. Whilst it is true to say that the use of monthly averages will impact how well resolved the snow to rain transition is in the model, the high uncertainty in the forcing data means the use of daily data will not represent a substantial improvement in representing this transition. Note the use of monthly averages for precipitation has been used previously for simulations using a forced atmosphere (e.g. Hunke and Bitz, 2009; Tsamados et al., 2015).

We have added the following discussion to the manuscript after the mention of the use of monthly averages for precipitation to address the above point:

'Monthly averages are often used for precipitation in atmosphere forced simulations (Hunke and Bitz, 2009; Tsamados et al., 2015) since there is a high uncertainty associated with precipitation in the reanalysis. The use of monthly averages over higher frequency forcing reduces this uncertainty, although there are limitations to this approach e.g. it will impact how well resolved the snow to rain transition is within the model.'

Line 149. "particularly in summer when there are lots of melt ponds"

Satellite estimate of ice thickness from altimetry are challenging because assumptions must be made about the snow cover on the sea ice. Please clarify how melt ponds result in uncertainty in satellite thickness estimates. And also include information about the challenges associated with snow on ice.

Satellite thickness products have historically not been produced in summer due to the presence of melt ponds (see e.g. Schröder et al., 2019).

We have modified the relevant text here to state more clearly that melt ponds have historically prevented the production of satellite thickness products in summer. We have also included a reference (Tilling et al., 2018) to highlight that snow on ice also presents an issue with confidence in ice thickness measurements.

Line 158. "The simulations show a slighter weaker sea ice trend"

Looking at the plot, it appears that the trend is perhaps half that observed which is quite large. Perhaps remove "slightly" (and note that there is a typo – "slighter" should be "slightly").

We have removed 'slightly' as suggested.

Figure 2 caption "between the 5 year study periods".

Does this mean that the differences shown are actually the 2010-2014 average minus the 1980-1984 average and the 2040-2044 average minus the 1980-1984 average? Please clarify.

Broadly yes, although averages are taken over the final 5 years of each decade (e.g. 2015-2019) rather than the first 5 years.

We have made the suggested clarification.

Line 162-163. "The interannual variability in the MIZ extent ... is large"

It appears that the MIZ variability is considerably smaller in the Bootstrap data. It may be useful to note this and also comment on the possible reason.

The differences between the SIC data products are primarily due to how the different products treat new ice and melt ponds, both of which will have a particularly large impact on the SIC in the MIZ. A full discussion of the different SIC data products and the reasons for differences between them is provided in Comiso et al. (2017).

We have made note of the substantial difference in MIZ variability between the two products. The updated text now reads:

'Figure 5 shows that whilst the total sea ice extent observations are generally in good agreement, satellite observations show a large range of estimates for the MIZ (see also Rolph et al. (2020)). The interannual variability in the MIZ extent in both simulations and observations in July, August and September (Figs 5b,c&d) is large, albeit smaller for the Bootstrap dataset than NASA Team.'

We have also added a reference to Comiso et al. (2017) earlier in the manuscript where we first introduce the satellite data products and note that this reference describes these two products in more detail and discusses the reasons for differences between them.

Line 165-166. "In 2010s the MIZ has become the dominant part of the sea ice cover in July, August and September."

Is this true for all the data? Or just the model runs? Please clarify.

This statement is broadly true for the two simulations and the NASA Team observations, but generally does not hold for the Bootstrap observations.

We have refined the statement to provide a more precise description of the applicability of the statement.

Line 173-174. Typo. There is a repeated sentence here.

Corrected as suggested.

Line 174-176. These few sentences basically repeat information from the earlier paragraph. Basically that model runs have an MIZ trend but observations do not. I'd suggest revising the two paragraphs (lines 160-176) to only present this information a single time. Having Figures 3 and 4 to illustrate this is fine but they can be discussed together instead of in separate paragraphs that have the same conclusions.

We have restructured and modified the relevant section (section 3.1 in the updated version) to discuss the two figures (now Figs 5 and 6) together and reduce the number of paragraphs from 3 to 2. Paragraphs 3 and 4 in section 3.1 now present a comparison of the total sea ice extent and MIZ extent respectively.

Line 205. "the MIZ is making up"

I'd suggest modifying to "the MIZ makes up"

Corrected as suggested.

Line 236. "slight decrease in top melt and a compensating increase in the fraction of basal and lateral melting"

Doesn't this statement depend on whether NCEP or HadGEM forced runs are being discussed? It looks like top melt increases in one (NCEP) but decreases in the other (HadGEM)? Please clarify

We agree with the referee's comment here, thank you for flagging this oversight.

We have clarified the relevant text to note that the primary change in the pack ice region is a substantial reduction in the fraction of ice volume lost to dynamics and corresponding increase in the fraction lost to thermodynamics, though there is also a redistribution between specific melt components e.g. there is a decrease in the fraction lost to top melt for the HadGEM-forced run, with a corresponding increase in fraction lost to lateral and basal melt.

Line 236-237. "It is surprising that there is not a more significant change in the becomes MIZ region over time in terms of proportion of melt that is top, basal, or lateral."

Could you please expand on why this might be the case and what this suggests?

The ratio of lateral to basal melt for a given floe is inversely proportional to floe size, so a lack of change in this ratio suggests limited changes in average floe size within the becomes MIZ region. This may partly be a result of how processes that drive the fragmentation of sea ice over the transition from pack ice to MIZ, in particular in-plane brittle failure, are currently represented in the FSTD model (Bateson et al., 2022). There is not a comparable simple geometrical relationship that can be used to understand any changes or lack thereof in the ratio of top to basal melt, as this ratio is impacted by any processes or properties that impact the surface energy imbalance (e.g. the elevation of the surface air and ocean temperatures above the freezing point). Isolating the impact of each individual process / property is non-trivial. Overall, the lack of change in the proportion of melt that is top, basal, or lateral within the becomes MIZ region over time shows that these melt ratios cannot be considered solely as a function of sea ice concentration.

We have included some of the above discussion points within the relevant paragraph to address the comment above.

Line 242. "There is much more top melting in both the always MIZ, and to a lesser degree the becomes MIZ region."

Should you reference Figure 8g and 8h here?

Thank you for this suggestion, although it is Fig. 9 being referred to here rather than Fig. 8.

We have now referenced panels 9g and 9h in the relevant location, as suggested. We have also modified the text to make sure it is clear we are referring to melt fraction, not total melt.

Line 253-254. "there is a strong decrease in the proportion of ice loss that is due to dynamics"

Could you clarify why this is the case? Presumably (and as said later) because the ice is thinner.

The above statement is correct, this will primarily be a result of thinner ice.

We have included the above clarification in the relevant location in the manuscript.

Line 261. "Top melting occurs first"

Really? It looks like basal melting occurs first or at a similar time, at least in the NCEP simulation. Do you mean instead that top melt peaks earlier? Please clarify.

Yes, this is supposed to refer to the peak melt rates.

We have modified the relevant statement to clarify that it refers to the peak melt rates.

Line 264. "Melting rates and growth rates"

Does this refer to the total melt/growth rates? Please clarify.

Yes, this does refer to the total melt/growth rates.

We have clarified the relevant statement to make it clear that it refers to the total melt/growth rates.

Line 273-274. "The difference is likely due to the greater summer surface air temperatures"

Since you are talking about the change in melt rates, wouldn't this be related to a change in the atmospheric forcing? And not a mean state difference? Please clarify.

The increase in melt rates that we see in the pack ice from the 1980s to the 2010s is not just specific to the period of peak melting, but across the entire melt season (particularly for the NCEP case). Hence the changes in melt rate that we see can be associated with the changes to the surface energy balance such as the increase in mean atmospheric surface temperature and downwelling longwave radiation shown in Fig. 2* (as is partly noted in the manuscript). These increases seem to be larger for NCEP than HadGEM2-ES over July to August i.e. the period where we see the peak melt rate, likely explaining the higher percentage.

*Figure 1 in the original manuscript.

We have clarified the relevant statement in the manuscript to include the above discussion points and make this point more clearly.

Line 275. "an increase in basal growth rates in the Autumn"

Why is there an increase in basal growth rates? Please clarify (and perhaps refer to earlier work which showed similar results).

The increased annual cycle is a combination of the change in surface energy balance (as shown in Fig. 2) and the relationship between ice thickness and ice growth rates (i.e. thinner ice grows more quickly due to faster rates of conduction). These two mechanisms combine to produce an enhanced annual cycle in sea ice. We acknowledge that the paper could be clearer in highlighting the separate role of these two mechanisms.

*Figure 1 in the original manuscript

We have modified the relevant section of the paper to better explain the separate effects (as described above) that combine to produce an enhanced annual cycle in sea ice volume.

Line 281-282. "whilst the end of summer melting ends later by 15 days..."

In the becomes MIZ region, doesn't the melt end earlier in 2010 and not later? It looks like the total melt curve is shifted earlier. Is this because all the ice melts out in this region? Please clarify and provide more explanations for the changes in budget terms where possible.

Thank you for identifying this error, the relevant sentence should read 'earlier' rather than 'later'. The shift in the total melt curve will be a result of multiple factors (e.g. changes to the surface energy balance), but the earlier peak and end of summer melting will be a result of reduced sea ice mass balance across the region, as suggested.

'Later has been corrected to earlier' in the relevant location. We have also included the above explanation for the earlier occurrence of the peak melt rate and end of summer melting in the relevant location.

Line 283-285. "The change in growth over October, November and December ..., with a larger increase in the NCEP simulation."

It looks like growth starts later (especially in NCEP) but then is higher in Nov and Dec? Perhaps clarify these details. And also discuss why this change occurs.

The delayed freeze-up will primarily be driven by changes in the surface energy balance (e.g. higher atmospheric surface temperatures) and an increase in heat accumulated over the prior melt season within the ocean surface mixed layer. The increase in November and December will again be a result of the inverse relationship between ice growth rates and thickness.

We have modified the relevant text to provide further discussion and explanation of the changes in growth rates, as above.

Line 289. "reflecting a significant reduction in summer ice in that region in the simulation."

Do you mean here that this reflects that ice has melted out?

Yes, this is what we are referring to here.

We have modified the relevant statement in the manuscript to note that this will be due to melt out of the sea ice.

Line 289-290. "Growth rates in both simulations decrease "

Why? Why not the increase seen in the other regions? Presumably this is related to ocean heat that is present and atmospheric warming that counteracts the ice thickness-growth rate relationship? Please provide more information on the reasons behind simulated mass budget changes.

Given that, both in the 1980s and 2010s, there will be very little to no sea ice across this region by the end of the melt season, there will be very limited benefit to the net growth rates resulting from the inverse relationship between ice thickness and growth rates. Hence other factors i.e. ocean heat content, atmospheric surface temperatures will be the primary drivers behind any differences in growth rates.

We have modified the relevant text to provide additional discussion as above.

Line 301. "there might be missing feedbacks"

This should be changed to "there are missing feedbacks".

At the very least, the air temperature does not respond to the sea ice state or surface fluxes which is clearly a missing feedback.

Corrected as suggested.

Line 301. "though it is possible this impact is not as large as previously assumed (Kay et al., 2016)"

The reference provided here only speaks to cloud interactions and in particular a lack of summer cloud feedback in response to sea ice loss. However, it does point out that fall clouds respond to ice loss. This is just one aspect of potential feedbacks and I do not believe that it suggests ice-atmosphere feedbacks are not as large as previously assumed (except perhaps for a summer cloud feedback). Please either clarify the Kay et al. study results (that your statement is specific to summer cloud feedbacks) or remove this statement.

We agree with the above comment regarding the limited applicability of the reference used here.

We have modified the text to better reflect the applicability of the reference being used here.

Line 302-303. "CMIP6 models fail to simulate a realistic"

Please qualify this statement as some CMIP6 models do have a reasonable ice area seasonal cycle.

We have updated the relevant statement to better reflect the findings of Notz and Community (2020):

'CMIP6 models underestimate the sensitivity of September sea-ice area to a given amount of global warming (Notz and Community., 2020).'

Line 303-304. "allows us to simulate a more realistic sea ice state"

The use of reanalysis-based forcing does not ensure a more realistic simulated ice state than that produced in a coupled simulation. For example, your NCEP and HadGEM forced runs have comparable biases (although often of different sign or in different seasons). I would suggest rephrasing.

We agree with the above comment. A reanalysis allows a more *accurate* simulation, in the sense of minimised difference compared to observations. A reanalysis does not necessarily allow a more *realistic* simulation, which is quantified more in terms of variability, trend, mean state etc. In particular, a reanalysis will be more likely to capture specific observed events than the equivalent climate forcing.

We have replaced 'realistic' with 'accurate' and have included the above explanation for using the term accurate within the manuscript.

Line 300-305. The motivation for using a forced sea ice simulation is discussed here. To me, it seems like a primary motivation here for the forced sea ice run framework was to allow for a more physics-rich model (with an FSTD) than is available in current coupled modeling systems. If this is true, I think that it is useful to state that here.

We agree, the above comment highlights a key advantage of using a forced sea ice simulation.

We have modified the relevant paragraph of the discussion section to make the point about physics highlighted above and better explain our motivation for using forced sea ice simulations

Line 317-318. "which may lack some ocean warming that we might expect to see in the 2040s"

Note that this would also affect sea ice growth.

We agree with the above comment.

We have modified the relevant text to note the impact on sea ice growth.

Figure 9. The values shown are the fraction of ice loss but the y-axis says fraction of melt. Please correct the y-axis label.

Thank you for flagging the above error.

We have corrected the relevant figure caption as suggested.

Line 322. "The differences in lateral melting was"

Please change "was" to "were"

Thank you for flagging the above typo.

Corrected as suggested.

Line 325-327. "The increase in top melting in the 2040s in the projection supports the importance of the topological melt pond scheme…"

You do not show anything specific to melt ponds in your results. As such it is unclear that there is an increased role for melt ponds or that any such increase is somehow specific to the particular melt pond scheme. Please rephrase this and clarify that it is speculation (if it is such. if not, please provide some results).

We acknowledge and agree that we have not presented any results that demonstrate the specific role of melt ponds here. However, there are other studies that explore the impact of different melt pond schemes that we can refer to here.

We have rephrased this sentence to note that whilst we have not explored the role of specific melt pond schemes in this study, comparable studies have explored and demonstrated the importance of the choice in melt pond scheme (see Diamond et al., 2024).

Line 334-336. "The MIZ and pack ice divide we have used differentiates between where internal stresses… and the sea ice is in free drift."

While this might be true, it appears to have little impact on the simulated results shown. It would be useful to clarify this.

Thank you for the above comment. Whilst Figs. 8-9 do show a greater importance for dynamics as a source of sea ice loss in the pack ice compared to the MIZ, it is not the primary reason for making this distinction.

We have removed the relevant comment and made corresponding modifications to the discussion section to better clarify the motivation for performing this analysis by partitioning the sea ice between pack ice and MIZ.

General comment for Results and Discussion section.

As mentioned in the specific comments, it would be helpful to provide more information on why certain changes are seen in the various mass budget terms and why this differs based on MIZ/non-MIZ regions. This should be provided in the results section but returned to within the discussion to clarify how the separation of analysis based on the MIZ provides new information/insight on the changing ice cover.

As highlighted in our responses to the specific comments, we now provide explanations for key changes in the mass budget terms and why these differ across the MIZ and pack ice within the results section. We now also include several paragraphs within the discussion section that review these differences and highlights both key insights that we have gained from our concentration-based approach to the analysis and the limitations of this approach for understanding the impacts of increased MIZ fraction on the future Arctic.

Line 362-363. "The opposite is true for lateral melting which made up … becoming comparable to top melting in MIZ defined region in the 1980s and 2010s."

Are there any implications from this for providing a better understanding of sea ice loss?

There are implications to changes in the partitioning of sea ice melt between lateral, top, and basal. Of particular relevance to sea ice loss is that the impact on albedo feedback is different: top melt results in the thinning of sea ice and albedo feedback via the production of melt ponds; whereas lateral melt reduces sea ice volume without directly impacting sea ice thickness and produces albedo feedback via the direct creation of open water. Therefore, top and lateral melt both contribute to albedo feedback via different mechanisms. Note that quantifying the overall impact of changes in surface albedo is non-trivial and beyond the scope of this paper.

We have included details within the discussion section about the implications of changes to the relative volume of top and lateral melt for the sea ice mass balance and albedo feedback.

Line 364-365. "However in the 2040s, there was an increase in top melting in all regions due to increasing surface air temperature."

The increase in top melt looks very small between 2010 and 2040 (Figure 8g and 8h) especially in the always MIZ region. Could you better clarify what you mean here?

Note that the comment referred to above was made on the basis of Fig. 9 rather than Fig. 8, where the contribution of top melt to total sea ice loss does increase relative to other terms.

Given the edits made elsewhere to the manuscript, we have modified the relevant statement to make the more salient point here that top melt is a greater contributor to sea ice loss within the MIZ in the 2040s compared to the 1980s (particularly compared to lateral melt).

Line 369-370. "In the regions of MIZ, from the 1980s to the 2010s, and for the 2010s and 2040s, we see the melt season shift earlier."

For the 1980s to 2010s, isn't this more the case for the Always Pack Ice and Becomes MIZ regions? It looks like the shift in melt season is actually quite small in the Always MIZ region. Please clarify

Note that the above refers to the onset of the melt season, which we will clarify. The analysis used to determine the shifts in timing of the melt season showed an earlier shift from the 1980s / 2010s to 2010s / 2040s of 5 – 9 days in all cases. The shifts are always larger for the Always Pack region compared to the Always MIZ region, but such comparisons are less clear for the Becomes MIZ region. However, beyond these specific metrics, Fig. 10 makes it clear that increases from the 1980s to 2010s in both melt fluxes during the early melt season and the peak melt flux were larger for the pack ice and regions that transitioned from pack ice to MIZ compared to regions that were always MIZ.

We have clarified the relevant statement to make clear that it refers to melt onset and have modified Fig. 10 to make it easier to see the differences between simulations. We have also added further details to the concluding remarks section and elsewhere to highlight the larger changes in the annual melt fluxes for the Always Pack and Becomes MIZ regions compared to the Always MIZ region.

Line 370-371. "end of summer melting shifts earlier in all the regions, particularly in the region that remains MIZ"

It looks like this is mostly true for the Becomes MIZ region? (and then the Always MIZ region) Please clarify.

The end of the summer melt ends about 2-3 weeks earlier for all regions except both Always Pack cases and Always MIZ for the NCEP case (where changes range from 4 days earlier 4 days later). The smaller changes within the Always Pack regions will result from the percentage reduction in sea ice volume being smaller in these regions than elsewhere. The small change for the NCEP Always MIZ region seems to be an exceptional case that would require a more detailed analysis to understand.

We have provided additional comments and discussion within the manuscript to better clarify changes in the end of summer melting.

**Referee 2**

Firstly, we would like to thank the referee for providing thorough and thoughtful comments on our manuscript. They have been invaluable in improving the manuscript.

*General comment about our response to referee 2: Referee 2 has made a substantial number of comments of a stylistic nature i.e. about how the paper is structured and written, rather than the scientific content of the paper. Whilst many of these comments provide constructive advice, we believe that several of these comments are a matter of opinion rather than a fixed rule on how a paper should be written and / or structured. Given this, and the volume of comments of this nature, we propose to address these comments collectively at the end of the response and highlight the changes we have made to the paper in response to these comments, however we will only adopt those suggestions that we believe will have a net positive result on the paper.*

Since its last iteration, significant changes have been made to improve the clarity of the manuscript, the quality of the figure, and the presentation of the result. For that, I congratulate the authors for the effort that they have placed into assessing the reviewer's comments.

However, the clarification of the results has raised concerns regarding the scientific content that were not apparent in my initial review. Those issues must be addressed by the authors by doing additional analysis and it should be either acknowledged or clarified within the manuscript.

Additionally, I believe that the manuscript still needs improvements regarding the writing and the structure. While the changes proposed in the review are probably not all necessary, I believe they would enhance the effectiveness of the message and the quality of the scientific communication.

I maintain my opinion that the reviewed paper deals with the scientifically important research question of the widening of the marginal ice zone and the transition towards the ice-free Arctic. However, I believe that major revisions are still required to reinforce the analysis and improve the text.

We appreciate the recognition of the improvement in quality of the manuscript since the original submission and are grateful to the referee for their role in this process. The remaining concerns raised above have been addressed in detail via our responses to the comments below.

*Scientific concerns*

1) I was expecting larger differences between the volume fluxes in the "always pack-ice", "become MIZ" and "always MIZ" regions, especially for the partitioning between top, basal and lateral melt (Figs. 8-10). It is mentioned in the text that "the partitioning of the melt between top, basal and lateral melting does differ substantially between the pack ice and MIZ", but I must admit that I was not convinced about that by looking at the figures. As far as I am concerned, the changes in the volume flux (Fig. 8) and fraction of volume flux (Fig. 9) between the "always pack" and the "always MIZ" region (e.g. Fig. 8a and Fig. 8c) are almost of the same order of magnitude as the changes in volume flux simulated due to variations in the atmospheric forcing (e.g. Fig. 8c and Fig. 8f).

The changes in volume flux between the different regions are significant. For example, Fig. 8a shows that for the NCEP-forced simulation we see the basal and lateral melt fluxes approximately double from the 1980s to 2010s for the always pack region. We also see an increase of over a third in the top melt flux. In Fig. 9a we see that the net effect of this change is an increase in the melt contribution to sea ice loss in the always pack region from about 70% to 85%, reducing the contribution of dynamics by about a half. The changes for the HadGEM2-ES case for the same period are qualitatively the same, though the magnitude of the change is smaller. In comparison, Fig. 8 shows that for the always MIZ region we see very little change in the lateral melt annual volume flux from the 1980s to 2010s, but we do see small reductions (of order 5 to 10%) in the top and basal melt annual volume flux. In this region, there will be competing effects of the reduced sea ice concentration versus increases in melt rate due to changes in surface energy balance. For lateral melt rate, reductions in the average floe size will also have an impact. In Fig. 9, we see only small changes in repartitioning between the melt components but a net increase overall in the melt contribution to sea ice loss. By the 2010s the contribution of dynamics to sea ice loss is just a couple of percent in the always MIZ region

The results also clearly support the statement 'the partitioning of the melt between top, basal and lateral melting does differ substantially between the pack ice and MIZ'. For example, in Fig. 9 we see that for the 'always pack' region both top and basal melt contributions to total sea ice loss are in the range 25% - 50% across the 1980s and 2010s for both simulations, whereas in the 'always MIZ' region the basal melt contribution to total sea ice loss exceeds 60%, with the top melt contribution around or below 20%.

We expect the differences in volume fluxes between the different regions to be comparable in order to the sensitivity to the atmospheric forcing. There is no physical reason to expect different orders of magnitude, if this were the case this would indicate a problem with the model. The driving force for sea ice change is the imbalance between the atmospheric and oceanic heat fluxes. The principal impact of the MIZ physics is to alter the way in which that heat flux imbalance leads to sea ice mass balance change i.e. extent vs thickness.

We have modified the results section to better highlight the differences in the volume fluxes and how they change between the different regions. In particular, we have provided a more detailed discussion and explanation to justify the point that the partitioning of the melt between top, basal and lateral melting does differ substantially between the pack ice and MIZ.

I suspect that it might be related to the use of fixed regions for the analysis that are only based on the July SIC (Fig. 6). I believe that larger, and more significant changes could be shown by computing volume flux in regions that evolve monthly. While I don't think that doing such an analysis would affect the main message of the study, I believe that it would show more contrasting and interesting results.

As noted above, the results are reasonable, understandable, and significant. Our focus is on the change within the MIZ, not the local spatial and temporal variability. The advantage of using fixed regions for analysis is it means we can think about changes in these regions more clearly in terms of sources and sinks of sea ice i.e. if the net volume flux for sea ice loss processes is greater than for sea ice gain processes, then the total volume of sea ice within the domain will reduce. This will not be true if calculations are based on regions that are not fixed but instead evolve monthly, which then complicates interpreting any changes in these regions. The July SIC was selected to define these regions because both melt fluxes and MIZ extent generally peak in

July, so the partitioning between MIZ and pack ice in this month is most relevant to understanding the changing roles of the MIZ and pack ice to thermodynamic sea ice loss.

We have included the above discussion and justification of our choice of methodology within the discussion section of the manuscript to better explain why we selected this methodology.

2) In my experience, the CICE's FSTD model has shown to be extremely sensitive to the wave forcing used. The realism of the simulated mean floe diameter (MFD) pattern can often be questionable (e.g. Roach et al. 2018, 2019), and FSTD has been hypothesized to be biased toward generating small floe size. This unresolved issue is likely related, among other factors, to an unphysical parametrization of the wave fracture.

The version of the FSTD model used in this study is different to the formulation used in Roach et al. (2019) – see section 2.1 or Bateson et al. (2022) for further details of these differences. Results presented in Chapter 7 of Bateson (2021) show a relatively low sensitivity in the total perimeter density of smaller floes to perturbations in the representation of waves breakup in the model for Arctic sea ice simulations. In particular, Figs. 7.6 – 7.8 (Figs. 7.6 and 7.8 have been reproduced here for easy reference*) in this chapter show that when the wave attenuation rate is reduced by a factor of 10 (which represents a substantial perturbation to the model), the change in the perimeter density distribution is small and changes to the sea ice mass balance are on average around 0.5% and always less than 1% in magnitude. Results presented in the same chapter show substantially larger sensitivities to perturbations in other model components e.g. the brittle fracture scheme, the size of newly formed floes. Note we anticipate that a similar set of sensitivity studies performed in the Antarctic would likely show a higher sensitivity to the representation of wave break-up in the model, though this is not of relevance for this study.

*Within these figures, *prog-stan* refers to a simulation using the standard formulation of the FSTD model (this is the version used within this study), *prog-nowb* refers to a simulation where wave break-up is not allowed to modify the FSTD, and *prog-morewb* refers to a simulation where the wave attenuation rate is reduced by a factor of 10.

Regarding the hypothesised tendency of the FSTD model to be biased towards small floes, as noted above, the model framework we use here is different to the formulation used in Roach et al. (2019). Bateson et al. (2022) presents a comparison of model output from the version of the FSTD model used in this study to observations. Furthermore, Wang et al. (2023) provides a comparison of output of both formulations of the FSTD model (i.e. Roach et al., 2019; and Bateson et al., 2022) to satellite derived observations of the FSD. This study shows that both versions of the FSTD model produce a greater fraction of smaller floes compared to observations, however this difference is substantially larger for the version of the FSTD model presented in Roach et al. (2019). It is suggested by Wang et al. (2023) that this difference may be due to errors with the observations (e.g. insufficient resolution to detect small floes) rather than just being down to an inadequate capturing of relevant physics in the FSTD model.

Additional explanation / motivation for using this approach is now provided in section 2.1 where the FSTD model is described, and a more complete discussion of both the limitations of this approach in accounting for the impacts of waves on FSD evolution and justification for using this approach are provided in the discussion section. This extended discussion addresses the above concerns and notes that the FSTD model used here has been compared against observations.

While this is not in the scope of the study, I believe it's important for the authors to acknowledge that the wave forcing has a major impact on the FSTD and, consequently, potential implications for the results, given that lateral melt depends on the MFD. I find the approach to handle the wave forcing in the study (i.e. prescribing a cyclic forcing from ERA-I) questionable.

First, the authors mention that "although the wave forcing fields do not have any trends, the propagation of the waves into the ice field does respond to the changes in the ice cover over time". However, if the wave forcing is prescribed at each grid point from a climatology, how does it consider the change in the ice cover over time? Is the wave forcing only prescribed at the ice edge with an attenuation computed inside the ice cover? Then, how does it deal with a grid point that was in the pack ice in the 2010s, but that is ice-free in the 2040s?

What wave properties are prescribed at that grid point then? Second, while the wave forcing is cyclic, the atmospheric forcing (including the winds) is not. If so, it does mean that after 2017 the wind pattern might not match with the wave pattern even if, in reality, those two concepts should be tightly interconnected.

While those two concerns suggest that the wave forcing used in the HadGEM2-ES projection is unrealistic, it might not be significant for the results as long as the MFD remains somewhat realistic. Therefore, for the sake of the discussion, it would be beneficial to look at your simulated annual cycle of the FSDRAD field (mean floe diameter) in the 1980s, 2010s, and 2040s.

The wave treatment described in Roach et al. (2018) is used here, rather than the modifications made in Roach et al. (2019). To determine the sea ice properties within a given grid cell, the model extrapolates along a line of longitude to the first ice-free grid cell where wave information is also present. Note that this does not inherently need to be at the ice edge, hence the model can easily adapt to changes in the sea ice edge.

As stated above, the FSTD model used in this study (i.e. including a representation of brittle fracture) has been shown to be realistic for the present-day Arctic in comparisons to observations by both Bateson et al. (2022) and Wang et al. (2023). The limited sensitivity to the treatment of wave break-up in this permutation of the FSTD model, as discussed above, should alleviate any concerns (such as the valid concern about the disconnect between the wind and wave pattern within the projections) regarding the extent to which the model is simulating a realistic wave field within the projections since it suggests that wave break-up will not be a substantial driver of change in FSTD behaviour from the 2010s to 2040s. The inclusion of a plot of the simulated annual cycle of a representative metric of floe size (such as the mean floe diameter, though other approaches exist) will not be illuminating in this regard since both the annual evolution of this metric and any changes over the time period will be driven not only by wave break-up but also, and to a greater extent, by the other processes included in the model, such as brittle fracture and floe welding, and it will be a non-trivial problem to isolate the specific impact of waves.

An explanation of how the model accounts for discrepancies between the simulated sea ice edge and wave forcing data is now provided in section 2.1. As discussed above, we have provided additional details within the model description and discussion to address the highlighted concerns.

[Figure]

**Figure 7.6:** The perimeter density distribution, $m\ km^{-2}$, of sea ice area as a function of floe size for April MIZ (top left), April pack ice (top right), August MIZ (bottom left), and August pack ice (bottom right). Distributions are shown for *prog-nowb* (red, cross, dashed), *prog-stan* (purple, diamond, dotted) and *prog-morewb* (blue, triangle, long-dash), all averaged over 2000 – 2016.

[Figure]

**Figure 7.8:** Difference in sea ice extent (solid) and volume (dashed) of *prog-nowb* (blue, diamond or triangle) and *prog-morewb* (red, cross or star) relative to *prog-stan* averaged over 2000 - 2016.

***Specific Comments***

*Abstract*

The abstract is the first part that most of the readers read and sometimes the only part that they will read. For this reason, it is probably the part of the manuscript that needs to be written with the utmost care. It needs to tell the story on its own while being as dense, concise, and as clear as possible.

Line 1 and 4-5: Repetitive sentence; the authors should remove one.

The two sentences here have different meanings; an increase in the MIZ fraction does not necessarily imply a wider MIZ (see, for example, Rolph et al., 2020).

Line 5: Typo ? pack ice.

Ice pack and pack ice are interchangeable terms.

Line 9: "The model has been compared to floe size distribution observations." Is this true? I have been reading the manuscript repeatedly and I do not see any comparison with floe size distribution observations. I only see comparisons with SIC and ice thickness observations.

Yes, the FSTD model used here has been compared to observations of the FSD in prior studies, specifically Bateson et al. (2022) and Wang et al. (2023).

We have modified the relevant sentence in the abstract to make it clearer that this comparison was made in previous studies:

'We validate this model using satellite observations of sea ice extent and PIOMAS estimates of thickness. A comparable setup has also been compared to floe size distribution observations in prior studies.'

*Introduction*

Line 23: applied is repeated twice. The authors should reformulate.

Agreed.

Line 26: "More fragile" feels vague. Thinner, more mobile, or more fractured could be more accurate.

We agree the language here could be more precise.

We have changed 'more fragile' to 'more vulnerable to breakup'.

Lines 36-40: The word "potentially" should be avoided. This has been the subject of many studies. Add references.

We disagree. 'Potentially' is correct here. Whilst an increase in lateral melt might be expected to enhance the ice-albedo feedback due to the formation of open water as noted in the manuscript, the size of the impact may be negligible, and other feedbacks also should be accounted for (e.g. indirect impacts on melt pond area). A feedback denial study would be required to properly quantify this effect.

We have added references to Bateson et al. (2020), Smith et al. (2022), and Curry et al. (1995) to the relevant section.

Lines 40-45: again, the use of "feedback" is repetitive and vague as it is not clearly explained here. Also, the word "action" feels vague. Wave-induced fracture could be more accurate. See "Floes, the marginal ice zone and coupled wave-sea-ice feedback" (Horvat, 2022), to improve that paragraph. Add references.

We have modified the relevant text to clarify the wave-floe size feedback mechanism. We have also added additional relevant references and details to this section including the Horvat (2022) reference.

Line 49: Add references.

We have included references to Flocco et al. (2012) and Rösel and Kaleschke (2012).

Lines 52-64: This is the "state the problem" section of the introduction. While other papers that made a sea-ice budget are cited, more emphasis should be placed on the lack of knowledge that persists after those studies. Instead, here, some of their results are stated, but they do not appear to be directly related to the content of this manuscript. For example, "They also found that the initial sea ice state was important in determining projected changes to the sea ice cover, with thicker initial ice resulting in more sea ice volume change." As far as I'm concerned, the manuscript does not investigate of the effect the initial sea-ice state on the volume budget, therefore the emphasize should not be placed on that. It should instead focus on things that are relevant to the findings of this study (e.g. do they compare their results to observations? Do they analyze the processes in the MIZ and in the pack ice separately? etc.)

We do not agree that the point about sensitivity to the initial sea ice state is not relevant to this manuscript, since this point highlights the need to ensure a realistic timeseries in sea ice extent and volume for studies evaluating the sea ice volume budget (given the broader implication here that the volume budget at a time t + x will be sensitive to the state of the sea ice at the prior time, t).

We have modified our comments here to highlight additional relevant findings from Holland et al. (2010) and to better clarify why sensitivity of volume change to initial sea ice state is a relevant and important point for this study. We have also now noted that this study evaluated the sea ice mass budget over a pan Arctic scale.

Line 69-70: The emphasis should be placed on the fact that CICE has been used in previous modelling studies that focused on the representation of the MIZ.

We have reformulated the relevant sentence to clarify that two of the three studies were focused on simulating MIZ extent (Rolph et al., 2020) or on improving the representation of MIZ processes (Bateson et al., 2020).

*Methodology*

Line 81: what is CPOM.

CPOM is the Centre for Polar Observation and Modelling.

We have now defined CPOM in the manuscript.

Lines 86-88: Also, I would be careful about saying that Roach et al. 2018,2019 give realistic simulations of the observed FSD as I do not think that there are any comparisons to observations in those studies.

Bateson et al. (2022) includes a comparison of FTSD model output to observations of mid-sized floes. Wang et al. (2023) also compared FSTD model output from both Roach et al. (2019) and Bateson et al. (2022) to observations of floe size, which we have discussed in more detail elsewhere (see response to scientific concerns).

We have modified the relevant text to clarify that a comparison to observations is made by Bateson et al. (2022).

Line 90: There is no need to mention the full name of the ocean reanalysis, it can be found in the reference (e.g. "a climatological ocean reanalysis (Ferry et al. 2011)").

We think that it is helpful to the reader to provide the full name of the ocean reanalysis here.

Lines 91-92: A separate sentence for the currents is unnecessary, it can be merged.

We would like to note that the original statement regarding the restoring of ocean currents was incorrect. The mixed-layer model used does not have an explicit representation of ocean currents, but instead uses restoring as a method to approximate the moderating effects of ocean currents on mixed layer properties. We apologise for this error.

We have corrected the relevant section.

Lines 106-109: I am not entirely convinced that the absence of trend reported in Bateson (2021) is a consensus. Many studies argue that the increase in open water in the summer that creates a larger fetch, combined with an increase in the intensity and frequency of storms in high latitudes will lead to waves of larger amplitude in the ice-free Arctic. The authors should read the following papers: Swell and sea in the Emerging Arctic Ocean (Thompson et al. 2014), Growth of Wave height with Retreating Ice Cover in the Arctic (Li et al. 2019), Sea Ice Retreat Contributes to Projected Increases in Extreme Arctic Ocean Surface Waves (Casas-Prat et al. 2020) and Wind and wave climate in the Arctic Ocean as observed by altimeters (Liu et al. 2016). It should be at least acknowledged in the paragraph that there is no consensus on that yet.

Bateson (2021) demonstrated, for the version of the FSTD model used here, a limited impact on the sea ice mass balance from reducing the attenuation rate of the waves propagating into the sea ice cover by a factor of 10. Whilst we agree that the available evidence base suggests that waves of a larger amplitude can be expected in a future Arctic, Bateson (2021) argues that the magnitude of this change is insufficient to drive major changes in the sea ice state on the basis of the sensitivity study presented.

We have modified the relevant section to better convey our argument here and have included some of the references highlighted above by the referee.

Lines 102-103: "The wave forcing consists of the significant wave height and peak wave period for the ocean surface waves". If the same FSTD model as in Roach et al. 2019, the CICE fracture model uses a wave spectrum to compute fracture, how is the wave spectrum derived from significant wave height and peak wave period only?

As noted in the manuscript, the model uses the fracture model outlined in Roach et al. (2018) rather than the version described in the 2019 paper. The latter is designed to use model output from separate wave-in-ice models e.g. WaveWatch III, whereas the former only required wave forcing external to the sea ice cover. For the 2018 version, lookup tables are computed prior to the simulation that calculate the fracture distribution for different combinations of input

parameters. These input parameters are significant wave height and peak wave period external to the sea ice cover and the number of floes and mean sea ice thickness between the edge of the sea ice cover and the present location.

We have modified section 2.1 of the manuscript to ensure the approach to wave modelling used in this study is clear.

Line 136: Add references.

We have added references here.

Fig. 1: The colors are confusing. The author should consider using a dashed line NCEP and a solid line for HadGEM (or vice-versa).

We have made the above change as suggested.

Line 138: If Rolph et al. (2020) have already done that comparison, why doing it again? The author needs to clearly highlight what is new about their paper and it must be clear from the "state the problem" part of the introduction.

Whilst Rolph et al. (2020) did make a similar comparison, this paper used an earlier version of the FSTD model to the one used here (it did not include the brittle fracture scheme) and that study only used the NCEP reanalysis for atmospheric forcing and not output from HadGEM2-ES. Hence, we cannot validate the simulations presented here solely on the basis of the results presented in Rolph et al. (2020).

We have modified the text to clarify the need for this comparison.

Lines 139-145: A basic description of the two SIC datasets used for validation should be there. How exactly is it measured? Is it both passive microwaves? Which satellite? Why is there so much difference between the datasets?

These are commonly used datasets for SIC observations, so it is unnecessary and beyond the scope of this paper to provide technical details of these products here. References have been clearly provided if readers do want more details about these SIC products. The differences between the SIC data products are primarily due to how the different products treat new ice and melt ponds, both of which will have a particularly large impact on the SIC in the MIZ. A full discussion of the different SIC data products and the reasons for differences between them is provided in Comiso et al. (2017).

We have made note of the substantial difference in MIZ variability between the two products and have included a reference to Comiso et al. (2017).

Line 145: "Daily values are used to compute monthly values of sea ice and MIZ extent that are plotted in Figure 3." This should be in the caption of the figure instead.

This sentence combines with the previous few sentences that describe how the SIC satellite data products have been processed for use in this study and therefore cannot be moved independently to the rest of this section.

Fig. 3: I am not sure of the necessity of: "Uncertainty levels of +/−10% were used for the satellite values in Rolph et al. (2020), they have been left out in these figures to make them clearer." as it is not discussed in the text nor shown in the figure.

Agreed, we have removed this comment.

Line 158: The sentence "slighter weaker sea-ice trend" is unclear. Is it really about the trend (i.e. the rate at which the sea-ice extent changes), or about the sea-ice extent as shown in the figure? Also, is it about the MIZ extent or the total sea-ice extent? The author can be more precise when referring to a figure (e.g. "slightly weaker MIZ extent trend (see thick line in Fig.3)").

We have clarified the statement, as suggested.

Line 164: "An increasing trend" in what? Again, the authors can be more precise in their statement.

We have clarified the statement, as suggested.

Line 184: Typo ? "for the scope of this study."

This is not a typo, but we agree the point here could be expressed differently.

We have modified the relevant text to improve clarity.

Line 198: This is already mentioned in line 141.

Yes, but we have reiterated this point to add clarity to the following discussion.

Lines 207-208: Unnecessary, it could be removed.

This sentence may not be strictly necessary, but we think it provides clarity to the reader on the structure of the manuscript.

*Results*

The description of the figures can often be confusing, especially for Figs. 8-10 which contains many panels, with many different lines and bars.

Fig. 8: It is unnecessary to repeat the "Volume flux" in the y-label three times in the subplot. I suggest putting one y-label for the whole subplot.

We believe that repetition of this y-label is important for clarity, given the format of the figure.

Fig. 9: "Fraction of the melt" in the y-label is inaccurate as the dynamic term is not melting. "Fraction of the total ice loss" would be more accurate. Again, the y-label does not need to be repeated. In the caption, I suggest the following phrasing: "The evaporation term is neglected."

We agree that 'fraction of the total ice loss' would be a better expression to use here, though we have used 'fraction of ice loss' for brevity. As above, we do not agree that that the y-label does not need to be repeated.

We have changed the y-label and figure caption as suggested.

Line 270: The threshold for the melting should be mentioned at the beginning of the paragraph. Is this threshold applied to the total melt or to one of the individual melt terms?

The threshold applies to the total melt.

We have added an additional paragraph prior to the one referenced to state the approach used to determine the different metrics used here such as melt onset, including a clarification that the threshold applies to total melt and growth.

Fig. 10: I believe that the y-label is incorrect. In Figure 8, the y-label is volume flux with units of m3/m2 and then in Figure 10 the y-label is also volume flux, but now with units of m3/m2 days. In the text, the author refers to growth and melt rates in lines 284-299 and I think that they are referring to Figure 10. I believe that the y-label should be growth/melting rate, but this needs to be clarified.

Volume flux in this study refers to the rate of volume loss or gain via the specified mechanism expressed as the equivalent change in ice thickness. In Fig. 8, this rate is expressed per year, whereas in Fig. 10, this rate is expressed per day. Growth/melting rate is an alternative way to express this concept, but it could also be applied to both figures. We do agree that there is potential for confusion here and some clarity is needed.

We have modified the y-label in Fig. 8 as discussed above.

Line 272: Which of the melting rates are you talking about? The total (black), the top (red), the lateral (purple) or the basal (yellow)?

Yes, we are referring to the total melt rate here.

In the new paragraph where we introduce this analysis, we make it clear we are referring to the total melt rate.

*Discussion and conclusion*

The material in the discussion is adequate. I believe that the two last paragraphs of the discussion could be improved by considering all the changes mentioned previously and by adding more direct references to the results or to previous literature (just like in the two first discussion paragraphs).

We have substantially modified the discussion section to reflect the edits made elsewhere in the manuscript and in response to the comments from both referees. As suggested, we have also added further references where appropriate to the discussion section.

In the conclusion, the author is doing a great job at summarizing their results. However, I believe that the authors are missing the opportunity to emphasize a clear take-home message. Which one of those results was unexpected, novel, or significant? Which unanswered questions were addressed in the manuscript? What are the key findings uncovered by the methodological framework? This ties back to my earlier comment in the introduction about finding the knowledge gap in the literature, which I believe is never clearly highlighted in the manuscript.

We have made edits throughout the manuscript, including the conclusion, to highlight the novelty of the approach of evaluating the sea ice mass budget for different regions of sea ice defined using sea ice concentration and now explain in more detail how this has enabled us to generate insights into the changing role of different processes to the sea ice mass budget and the implications of an increasing MIZ fraction for the future Arctic sea ice. We have also modified the final paragraph of the conclusion (now the penultimate paragraph) to highlight the key findings from our analysis:

'Our analysis demonstrates that a different balance of processes controls the volume budget of the MIZ versus the pack ice. In addition, we find the general shift towards a state of an earlier melt season and stronger peak melt rates to be larger for the pack ice and regions that transition from pack ice to MIZ compared to regions that are MIZ across the relevant time period. However, we find that the balance of processes in the 2040s cannot be understood solely through

changes in sea ice concentration; the surface energy balance remains a strong control on sea ice mass balance and also has to be accounted for. The processes controlling evolution of the MIZ in the 2040s are different from those controlling evolution of the MIZ in the 1980s and 2010s. This has substantial implications for the set of processes that need to be represented with higher physical fidelity in climate models in order to best capture the behaviour of the sea ice in a future Arctic e.g. melt ponds.'

Line 323: "The differences in lateral melting were very likely caused by the inclusion of the FSTD model (Roach et al., 2018, 2019; Bateson et al., 2022), although we did not directly test this within this study." I believe that your methodological framework allows to investigate such a question for example, by comparing the mean floe diameter in the MIZ and pack ice regions. My concerns about the wave forcing should be addressed or acknowledged here.

Whilst a plot comparing mean floe size diameter would be interesting in its own merit, the version of the FSTD model used here is explored extensively in chapter 7 and 8 of Bateson (2020) and Bateson et al. (2022), and as such we would prefer to maintain a focus in this study on the broader changes in the sea ice cover and avoid a specific focus on the FSTD. Nevertheless, we can refer to model output to inform our discussion here and provide a more definitive statement on the impact of floe size on regional differences in lateral melting.

We have modified the relevant section to provide a more definitive statement on the impact of the FSTD model on regional differences in lateral melting. As suggested, we have also discussed and addressed the concerns regarding the wave forcing here, including references to the studies that have compared the version of the FSTD model used in this study to observations.

Lines 331-338: While the other paragraphs bring interesting discussion points (e.g. the use of a coupled climate model vs a forced sea-ice model, the absence of warming trend in the oceanic forcing, the effects of using an FSTD and a melt pond scheme), I am struggling to understand what is the point that the author is trying to make in that paragraph. For example, with the sentence: "Our approach also has the advantage of simplicity, the more concentration categories the MIZ is split into, the more complex the analysis becomes, and the less clear the results." This appears to be a simple general statement that does not integrate or extend any of the results.

The intention of this paragraph is to provide some discussion of both the advantages and limitations of the selected methodology. As noted by the referee, there were other approaches that could be used for this sort of analysis, and by evaluating the choice of methodology in the discussion, it provides more complete context for the results presented.

We have modified and restructured this paragraph to ensure that the discussion in this paragraph more clearly relates to the rest of the results and discussion.

Line 370: Typo? "particularly in the region that is remains MIZ."

Thank you for spotting this typo.

Typo has been corrected.

Line 374: Typo? "Our analysis demonstrates a different balance of processes control the volume budget…"

Thank you for spotting this typo.

Typo has been corrected.

Line 374-378: This is a rather vague and general way of closing the paper. I believe that a stronger concluding remark could be made by including future research directions.

We have modified both the existing final paragraph to better highlight the key findings of this study.

**Response to referee 2 on comments relating to paper style and structure**

As mentioned earlier, there are a substantial number of comments regarding the structure and style of the paper and not its scientific or technical content. Rather than addressing these comments individually, we will instead summarise how we have modified the paper in response to these comments. We have adopted all those suggestions that we believe will improve the quality of the manuscript:

- We have adapted the suggested document organisation.
- We have reduced the abstract to a single paragraph.
- We have added a final statement to the abstract to highlight the principal conclusion of the study.
- We have included additional references in the introduction, particularly between lines 35 – 51.
- We have moved the sentence started in line 30, as suggested.
- We have moved lines 32 – 35 earlier in the introduction, as suggested.
- We have rephrased lines 38 – 40 to avoid repetition of the phrase 'ice-albedo feedback'.
- We have removed 'the first to our knowledge' from line 65, and instead added some additional discussion both here and elsewhere in the manuscript to better motivate and highlight the novelty of this study.
- We have moved the discussion about the FSTD model from the methodology in lines 86-88 to the suggested location in the introduction.
- We have created a new paragraph to describe the wave forcing in the methodology and have modified and restructured this paragraph to reflect edits made elsewhere.
- We have started a new paragraph where the spin-up is discussed (lines 110 – 113).
- Sections 2.4 and 2.5 have been combined to produce a new section called 'model validation'. This is section 3.1 in the updated manuscript.
- We have moved lines 200 – 206 to the new model validation section, along with Fig. 7.
- We have replaced Figure with Fig. where appropriate as per the journal style guide.
- We have made additional references to figures in the text where necessary to ensure clarity.
- We have introduced additional paragraphs into the discussion, as suggested.

***General comments***

While short sentences are typically preferred in a scientific document, I believe that the author might have pushed the limit a bit far in some sections. Reading the manuscript often feels a little bit robotic. I suggest trying to combine some of the short sentences to introduce variation in the sentence length throughout the manuscript and to enhance the rhythm of the paragraphs.

The organization of the document is essential; the readers want to know where to quickly find the information they are looking for. Currently, results are spread between the methodology (section 2) and the results (section 3), which makes the story hard to follow.

The methodology should be strictly about describing the data and the method used, so the reader has enough information to reproduce the results. This is not where you typically analyze data, even if it is for model validation. Therefore, I recommend creating a new result section called "Model validation" and moving Figs. 1,2,3,4,5,7, as well as their associated discussion to this new result section.

Here is a suggestion for a preferred document organization:

1. Introduction
2. Data and methods: this should include what is currently sections 2.1, 2.2, 2.3 and 2.6.
   2.1. Model set-up and forcings or Model configurations. Include the content of what is currently sections 2.1 and 2.2.
   2.2. Model validation data or Observational data. Include the content of the current sections 2.3.
   2.3. Analysis method or methodology Include the content of the current section 2.6. Move lines 200-206 to the results.
3. Results: this should include what is currently sections 2.4, 2.5, 3.1 and 3.2. The author should also consider having shorter and more evocative subsection title.
   3.1. Model validation. Include what is currently sections 2.4, 2.5 and the part of 2.6 mentioned previously and the discussion about Figs. 3,4,5,7.
   3.2. Annual total volume fluxes.
   3.3. Annual cycle of the main melt and growth terms.
4. Discussion
5. Conclusion

The length of the manuscript also increased a lot (i.e. it went from 15 pages to 21 pages). I believe that there are some lengths and redundancies, which can undermine the message. In the specific comments, I go through each of the sections of the manuscript, and I suggest changes and rephrasing to improve the conciseness and clarity.

***Specific Comments***

Lines 5-10: the method of the study is summarized, however, it is not mentioned that two atmospheric forcing (NCEP/HadGEM2-ES) are used. Then, in lines 11-13 when the discussion of the results starts with a comparison between these two atmospheric forcings. In short, the part of the abstract that talks about the method should directly lead to the results.

Line 10: The abstract usually only has one paragraph.

The last part of the abstract lists some of the results, but it should be instead focused on the principal conclusion of the study (i.e. the take-home message).

I believe that the reorganization proposed in the general comment will help to really highlight the story of the paper. Here is an example of how that structure could be used the convey the message more clearly in the abstract.

"The marginal ice zone (MIZ), defined as the region of the ice cover that is influenced by waves, is projected to become a larger percentage of the summer ice cover as the Arctic transitions to ice-free summers. Here, we compare individual processes of ice volume balance in the pack ice to those in the MIZ to establish and contrast their relative importance and examine how these processes change as the summer MIZ fraction increases over time. We use CICE, a physics-rich sea ice-mixed layer model forced with two atmospheric datasets; the HadGEM2-ES simulation

(1980-2050), to simulate the ice cover in a high emission global warming scenario and the NCEP reanalysis (1979-2020), for comparison during the observational period. First, we compare both simulations to satellite-derived sea ice concentration (i.e. NASA Team/Bootstrap) and PIOMAS estimates of sea-ice thickness. Results show that [results from section 3.1] Then, we compare the annual volume fluxes for the following periods: low MIZ (1980s), high MIZ (2010s), and all MIZ (2040s), showing that [results from section 3.2]. Finally, we look at the annual cycle of the main melt and growth terms [results from section 3.3]. Those results highlight that [the take-home message]."

The introduction does a decent job of stating the background information and covering existing literature, but I still believe that it could use some rewriting to enhance the quality of the scientific communication. Here is what I suggest the authors to keep in mind to improve the flow of the introduction.

First, "contextualize the background information"; it needs to go from the most general (i.e. general statement about sea-ice, the MIZ, global warming, etc.) to the most precise (i.e. things that are directly related to the paper, e.g. the processes affecting the sea-ice volume budget). Second, "state the problem"; this is where similar previous literature is cited, and the knowledge gap or general misunderstanding of the problem is stated. Third, "address the problem"; this is where the methods, the scope of the paper, and what is unique about it are mentioned.

Additionally, some statements are made without citation; the introduction could use additional references especially between lines 35-51.

Line 7: "This is the first study (to our knowledge) that separately considers the pack ice and MIZ in this way." This type of sentence should be avoided; you either know that this is the first study, or you don't. The authors should conduct the appropriate literature review to make sure that it is or remove the sentence.

Line 30: This is still the "contextualize the background information" part. I suggest moving the sentence to the "address the problem" part.

Lines 32-35: This is a general statement about sea ice. I suggest moving this part to the beginning where the different SIC threshold for the MIZ definition (i.e. lines 22-25) is discussed.

Lines 38-40: the use of "ice-albedo feedback" is repetitive. The sentences could be merged or rephrased to avoid repetition.

Line 65: This is the "addresses the problem" part of the introduction. Again, I suggest changing the phrasing "the first to our knowledge". That could be implied by improving the previous paragraphs.

Line 71-77: A grocery list to describe the content of the manuscript should be avoided. The author should instead make a story out of it or remove it. This especially applies to the last part which simply states that section 4 is called discussion and section 5 is called conclusion. I think believe that the structural change proposed in the general comment will help to make a story out of the results. Moreover, a similar grocery list is made at the end of the methodology (lines 207-209), I suggest keeping only one.

As I mentioned before, the methodology should not contain any figures. The only figure that could potentially stay in this section is Fig. 6, as it serves more as a mean of better understanding the methodology rather than a result.

I recommend moving Figs. 1 & 2 to the appendix as well as the discussion of those figures, to reduce the length of the manuscript.

Moreover, the section is to be called "Data and methods", as this is also where the datasets used in this study are described.

Lines 86-88: "found to give realistic simulations of observed floe size distributions (FSD) for mid-range floe sizes in the Arctic. This model, minus the brittle fracture addition to the FSTD model, has been used previously by Rolph et al. (2020) to compare changes in the MIZ in a sea ice model to satellite observations." This should be mentioned in the introduction, especially where it is argued that CICE is a good model to represent the MIZ (i.e. lines 68-10). The method should instead only focus on describing the component of the model used.

Lines 93-98: This paragraph can be further simplified especially if the namelist is given in the supplementary material or in the code availability section. For example: "We use a number of the default CICE settings, including the layers thermodynamics of Bitz and Lipscomb (1999), Maykut and Untersteiner (1971) conductivity, Rothrock (1975) ridging scheme, the delta-Eddington radiation scheme (Briegleb and Light, 2007), and the linear remapping ice thickness distribution (ITD) approximation (Lipscomb and Hunke, 2004). Additionally, we use a prognostic melt pond model (Flocco et al., 2010, 2012) and an anisotropic plastic rheology (Heorton et al., 2018; Tsamados et al.,2014; Wilchinsky and Feltham, 2006)."

Lines 98-114: This deserves a new paragraph; this section might need to be reworked according to the answer to the second scientific concern comment. Additionally, the authors should reorganize the paragraph by first describing the way that the wave forcing is dealt with in their own study. Then, they can briefly mention how others have done this differently and why they argue that their method is adequate. For example, the paragraph should start with:

"The wave forcing used in this study is prescribed from on ERA-I reanalysis wave data and is repeated after 2017 (Bateson et al. 2022). The wave properties used are the significant wave height and the peak wave period, which are then extrapolated and updated every 6 hours in grid cells that contains less than 1% sea ice (Roach et al. 2018). This wave forcing set-up then differs from Roach et al. (2019) [...]"

Line 110: I suggest making a new paragraph for the discussion of the spin-up (i.e. lines 110-113) or moving this section after the discussion on the component of the model (line 99).

Line 115: At this point, the author should have already made clear that they are comparing the result from two atmospheric forcing before as this is a key part of the story. It needs to be mentioned in the abstract and the "address the problem" part of the introduction.

Figs. 1 & 2 are complementary to the main findings, but they are not essential for the story. I suggest moving them to an appendix with lines 124-136. The figure can be referred to later, when NCEP and HadGEM simulations are compared in the results. Lines 115-123 can be moved to the "Model configuration" section.

Lines 146-151: Most of it can be cut as it is general sea-ice knowledge. The paragraph could be simplified as: "To validate our simulated sea-ice thickness, we use the Pan-Arctic Ice Ocean

Modeling and Assimilation System (PIOMAS) (Zhang and Rothrock, 2003), a model that assimilates a range of sea ice area/concentration observations to give an estimate of continuous Arctic Sea ice volume changes over time. As with the satellite data, PIOMAS has been interpolated on the ORCA tripolar 1∘ grid and the CICE land mask has been applied."

Lines 153-155: Most of it is already in the caption of Figure 3, it can be deleted and rephrased.

Lines 158-159: "Whilst" is used in two subsequent sentences. Rephrase.

Lines 178-179: Unnecessary, this is already mentioned in the caption. Also, this part does not require a separate section, just add that paragraph to the previous section and call it "model validation".

Lines 187-189: Again, this is unnecessary as it is already mentioned in the caption of the figure. This comment applies to most of the figures. For example, the paragraph could be simplified as: "We consider three different ice cover states within the simulations: a low MIZ state in the 1980s; a high MIZ state in the 2010s; and an all MIZ state in the 2040s. In each case we use the last 5 years of daily July SIC and assign each grid cell as pack ice (SIC≥80%), MIZ (15% ≤SIC<80%) or open water (SIC<15%) (Fig. 6). The authors should avoid breaking the flow of the paragraph with sentences like "Figure 6 shows ... [content of the caption of Fig. 6].". This is repetitive as the reader will already look at the caption to understand what is in the figure.

Lines 200-206: This should also be moved to the new "model validation" results section as well as Fig. 7.

The caption of Figure 6 can be simplified. Change "Region 1 (blue) indicates the area that is pack ice in both the 1980s(2010s) and 2010s(2040s) in subplots a-d(e). Region 2 (green) indicates the area that is pack ice in the 1980s(2010s) and becomes MIZ in the 2010s(2040s) in subplots a-d(e). Region 3 (orange) indicates the area that is MIZ in both the 1980s(2010s) and the 2010(2040s) in subplots a-d(e)." to "Region 1 (blue) indicates the area that is pack ice, region 2 (green) indicates the area that is pack ice and becomes MIZ and region 3 (orange) indicates the area that is MIZ". Identify the period in the title of the subplot (e.g. (a) Bootstrap 1980s-2010; (b) NASA Team 1980s-2010; (c)... etc.)

The figures and the description of results are much clearer than in the last iteration and the results paragraphs are straightforward. However, as I mentioned previously, the author should avoid repeating the content of the caption in the text when referring to a figure.

Also, when referring to a figure, direct and indirect citations are possible. The direct method is to directly describe the figure in the text. For example.

"The annual volume fluxes for sea ice processes are shown in Figure 8.",

The indirect method is the cite the figure in parentheses at the end of the sentence that describes the results.

"Analysis annual volume fluxes for sea ice processes show that congelation growth dominates sea ice growth [...] (Fig. 8)."

More indirect references should be made throughout the text, especially for figures that do not require a lot of explanation, as it is less wordy. This will also help to avoid repeating the content of the caption. Also, while using the indirect approach use the abbreviation Fig. instead of Figure to lighten the text.

Finally, the author should consider the opportunity of using the indirect approach to add additional and more detailed references to the figures throughout the result section.

Line 257: "The same regions and time periods defined in Figure 6 have been used." Already mentioned in the caption.

Lines 278-298: Are those results related to Figure 10? If so, refer to the figure in the text, at least one reference per paragraph. For example, "In the always pack region peak melt increases and the melt season gets longer in the 2010s relative to the 1980s by 13 days in the NCEP and 6 days in the HadGEM2-ES forced simulations (Fig. 10a,d)."

Line 305: New paragraph?

Line 325: New paragraph?

Lines 343-344: Rephrase.

Lines 344-345: "The MIZ is defined as having a sea ice concentration (SIC) between 15% and 80% and pack ice is defined as SIC>80%." I believe that this should be included in the first sentence of the conclusion.

**References**

*We only include additional references here that are not already provided in the manuscript.*

Giles, Katharine A., Seymour W. Laxon, Andy L. Ridout, Duncan J. Wingham, and Sheldon Bacon. "Western Arctic Ocean freshwater storage increased by wind-driven spin-up of the Beaufort Gyre." *Nature Geoscience* 5, no. 3 (2012): 194-197.

---

## Author Response (AR3)

**Response to Referee Comments**

Referee comments are shown in black, our response is in blue, and changes to the manuscript are shown in red.

**Referee 1**

The authors have made substantial changes with considerable improvements in the manuscript. I just have a few minor comments that should be considered.

Firstly, we would like to thank the referee for reviewing the manuscript again and for the thorough and thoughtful comments on our manuscript that they have provided during the peer review process. We appreciate the recognition of the improvement in quality of the manuscript. We have addressed the remaining comments as detailed below.

**Minor comments.**

Abstract, line 20. "shifts 14 days earlier"

I'd suggest that you explicitly state here what time period this is relative to.

**Edit made as suggested.**

Line 291. "no clear change over time"

Later in the paragraph, you mention that there is a clear change over time (with an increase in growth in the "always pack" region from the 1980s to the 2010s). Please either remove or revise this "no clear change over time" statement.

Thank you for identifying the above oversight. The comment has been removed, as suggested.

Line 304. "Figure 8 shows a large reduction in congelation growth"

I believe that this is referring to the change in 2040. This change looks quite small, especially in the "becomes MIZ" region. Perhaps remove the word "large" here or revise in some other way.

Agreed, the word 'large' has been removed as suggested.

Line 455-456 "it does not account for any ocean warming that we might expect to see in the 2040s"

This statement could be confusing since ocean warming is possible in your simulations from changes in surface fluxes. This is stated further in the paragraph. To clarify for the reader, I'd suggest revising this statement to better clarify that it is specific to ocean warming associated with changing ocean heat flux convergence.

A caveat to address the above point has been added to the text:

'it does not account for any ocean warming that we might expect to see in the 2040s beyond that captured in the model via changes to atmospheric surface fluxes'

Line 485-486. "There is no significant difference over time or between the pack ice and MIZ in the processes of sea ice growth."

This statement seems misleading since you show that there is a difference between 1980 and 2010 in the congelation growth in the "always pack ice" region. Please revise.

Thank you for identifying the above oversight. The sentence referred to here has been replaced with the following:

'The main difference in the growth terms was a general increase in these terms for the 'always pack' region from the 1980s to the 2010s, likely resulting from the higher growth rates associated with thinner ice.'

**Referee 2**

**Summary:**

This is a modeling study striving to shed light on changes in sea ice volume melt as the Arctic sea ice cover shifts from a more compact state to a state with a more open ice cover for which processes such as lateral melt or advection into warmer water can play an enhanced role. The authors use an improved version of CICE forced by two different atmospheric models (one for present day and one for a seamless investigation of present day conditions and future projections) for their study. They do a consistency check of the model forcing by means of discussing the forcing itself and by comparing resulting model sea ice quantities such as timeseries of the total sea ice extent and volume against independent data. They investigate and discuss the temporal changes of ice volume melt for three regions defined by means of different temporal development of their characteristic sea ice concentration. They investigate processes such as ice growth, ice melt (top, basal and lateral), and dynamic processes and provide a quite sound discussion of their results.

I have mixed feelings with this manuscript. On the one hand it seems to be well written and many aspects of the study are laid out very well, illustrations of results are quite comprehensive as is the discussion. However, I have a few question about why certains things were done as they were done; I am not sure whether dynamic processes have been investigated as thorough as it might have been needed (also in terms how good the atmospheric forcing is in this regard); I am also not sure whether the findings are relevant in view of the substantial differences in the forcings of the two models used for the atmospheric forcing and I hence do not have a clear feeling about the uncertainty of the results shown. Finally, I find some expressions and definitions as written could lead to mis-understandings and mis-interpretations of the results shown and discussed.

Firstly, we would like to thank the referee for providing thorough and thoughtful comments on our manuscript. They have been invaluable in improving the manuscript. We have addressed the concerns raised above in our response to specific comments.

**General Comments**

GC1: I can understand that you applied the easier-to-use definition of the MIZ. My question is, however: How close is this definition to the one based on ocean waves and what would be the difference you'd expect in case you could use the wave-based definition? I am concerned about that. I thought that, in the meantime, the community has moved away from defining the MIZ as the area with sea-ice concentrations below 80%. It has been shown that this sea-ice concentration threshold is not adequate to approximate the part of the sea ice cover that is

influenced by waves. I was wondering whether the strength of the sea ice cover, i.e. discriminating between compact pack ice and freely drifting ice wouldn't be a more natural way to define the two different sea ice areas used in your study. I would avoid to term this MIZ or marginal ice zone given the vast extent the MIZ as defined by you has.

Firstly, both the concentration-based and wave-based definition of the MIZ continue to be used in the literature. It is not true to say that the community has moved away from using the concentration-based definition (see e.g. Concetta et al., 2024; Strong et al., 2024; and our own previous papers on this topic). We think it would be misleading to the community to avoid the use of the term MIZ when considering a concentration-based definition, as this would be going against several decades of accepted usage. We apply the concentration-based definition in this study since waves are not the critical factor in determining sea ice mass balance in the Arctic (Bateson et al., 2020; Bateson, 2021). We agree that the wave-based and concentration-based definitions of the MIZ do likely refer to different regions of the sea ice (Horvat et al., 2020). It is therefore important to be clear in the definition of the MIZ being used in this study, and this is indeed something that we have stated clearly in the opening few sentences of the introduction.

We have updated the text with the more recent references included above. We have included an additional sentence in the first paragraph of the introduction to address some of the above concerns:

'Whilst the two definitions of the MIZ likely refer to different regions of the sea ice (Horvat et al., 2020), we apply the concentration-based definition in this study since waves are not the critical factor in determining sea ice mass balance in the Arctic (Bateson et al., 2020; Bateson, 2021).'

GC2: Closely connected to GC1 is the question: Did you check how your results would change if you would change the threshold from 80% to 60% or to 90% - independent of the ice strength definition by Hibler?

A threshold of 80% (corresponding to the upper limit of the concentration-defined MIZ) is both a well-established threshold for distinguishing between regions of sea ice in the literature (as noted above and in the manuscript) and is physically motivated since it approximately corresponds to the transition from ice in free drift to pack ice. We have previously considered applying different thresholds, however there are also practical reasons why we selected a threshold of 80%, in addition to the physical motivation and widely-accepted usage. If we increase the threshold to 90%, much of the central pack region will be identified as MIZ due to the opening of leads. If we reduce the threshold to 60% or even 70%, the 'always MIZ' region over the 1980s to 2010s period will be too small to allow any useful analysis. Figure 6 already shows the limited extent of this region even for a threshold of 80% (see the orange areas in panels c and d).

We have included the following comments in section 2.3 of the manuscript to provide more details of our choice of threshold:

'We selected a threshold of 80% for our analysis because it is both a well-established threshold for distinguishing between regions of sea ice in the literature (i.e. the upper limit of the concentration-defined MIZ) and is physically motivated since it approximately corresponds to the transition from ice in free drift to pack ice. In additional, alternative choices for this threshold have practical limitations e.g. for a higher threshold of 90%, much of the central pack region will be identified as MIZ due to the opening of leads. For a lower threshold e.g. 70%, too few grid cells are identified as part of the MIZ in the 1980s to allow useful analysis.' GC3: I am aware that you have expanded the work put into the initial version of this manuscript substantially. But given the importance of surface turbulent heat fluxes and the potential increasing role of sea ice transport from one region to another region, I was wondering whether you should not comment and/or justify why you did not also check NCEP2 and HadGEM2-ES model data with respect to wind speed and direction